# Particle Dual Averaging: Optimization of Mean Field Neural Network with Global Convergence Rate Analysis

**Atsushi Nitanda**
Kyushu Institute of Technology
RIKEN Center for Advanced Intelligence Project
nitanda@ai.kyutech.ac.jp

**Denny Wu**
University of Toronto
Vector Institute for Artificial Intelligence
dennywu@cs.toronto.edu

**Taiji Suzuki**
University of Tokyo
RIKEN Center for Advanced Intelligence Project
taiji@mist.i.u-tokyo.ac.jp

## Abstract

We propose the *particle dual averaging* (PDA) method, which generalizes the dual averaging method in convex optimization to the optimization over probability distributions with quantitative runtime guarantee. The algorithm consists of an inner loop and outer loop: the inner loop utilizes the Langevin algorithm to approximately solve for a stationary distribution, which is then optimized in the outer loop. The method can thus be interpreted as an extension of the Langevin algorithm to naturally handle *nonlinear* functional on the probability space. An important application of the proposed method is the optimization of neural network in the *mean field* regime, which is theoretically attractive due to the presence of nonlinear feature learning, but quantitative convergence rate can be challenging to obtain. By adapting finite-dimensional convex optimization theory into the space of measures, we analyze PDA in regularized empirical / expected risk minimization, and establish *quantitative* global convergence in learning two-layer mean field neural networks under more general settings. Our theoretical results are supported by numerical simulations on neural networks with reasonable size.

## 1 Introduction

Gradient-based optimization can achieve vanishing training error on neural networks, despite the apparent non-convex landscape. Among various works that explains the global convergence, one common ingredient is to utilize overparameterization to translate the training dynamics into function spaces, and then exploit the convexity of the loss function with respect to the function. Such endeavors usually consider models in one of the two categories: the *mean field* regime or the *kernel* regime.

On one hand, analysis in the kernel (lazy) regime connects gradient descent on wide neural network to kernel regression with respect to the neural tangent kernel (Jacot et al., 2018), which leads to global convergence at linear rate (Du et al., 2019; Allen-Zhu et al., 2019; Zou et al., 2020). However, key to the analysis is the *linearization* of the training dynamics, which requires appropriate scaling of the model such that distance traveled by the parameters vanishes (Chizat and Bach, 2018a). Such regime thus fails to explain the *feature learning* of neural networks (Yang and Hu, 2020), which is believed to be an important advantage of deep learning; indeed, it has been shown that deep learning can outperform kernel models due to this adaptivity (Suzuki, 2018; Ghorbani et al., 2019a).

In contrast, the mean field regime describes the gradient descent dynamics as Wasserstein gradient flow in the probability space (Nitanda and Suzuki, 2017; Mei et al., 2018; Chizat and Bach, 2018b),

35th Conference on Neural Information Processing Systems (NeurIPS 2021).

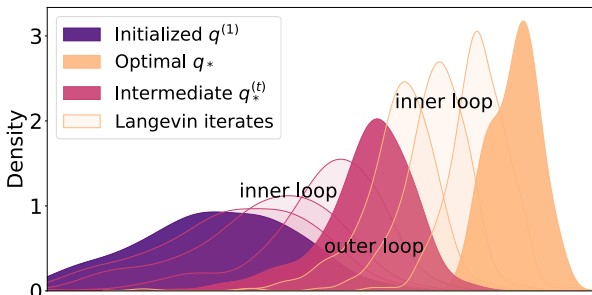

Figure 1: 1D visualization of parameter distribution of mean field two-layer neural network (tanh) optimized by PDA. The *inner loop* uses the Langevin algorithm to solve an approximate stationary distribution $q_*^{(t)}$, which is then optimized in the *outer loop* towards the true target $q_*$.

which captures the potentially *nonlinear* evolution of parameters travelling beyond the kernel regime. While the mean field limit is appealing due to the presence of "feature learning", its characterization is more challenging and quantitative analysis is largely lacking. Recent works established convergence rate in continuous time under modified dynamics (Rotskoff et al., 2019), strong assumptions on the target function (Javanmard et al., 2019), or regularized objective (Hu et al., 2019), but such result can be fragile in the discrete-time or finite-particle setting — in fact, the discretization error often scales exponentially with the time horizon or dimensionality, which limits the applicability of the theory. Hence, an important research problem that we aim to address is

*Can we develop optimization algorithms for neural networks in the mean field regime with more accurate quantitative guarantees the kernel regime enjoys?*

We address this question by introducing the *particle dual averaging* (PDA) method, which globally optimizes an entropic regularized nonlinear functional. For two-layer mean field network which is an important application, we establish polynomial runtime guarantee for the *discrete-time* algorithm; to our knowledge this is the first quantitative global convergence result under similar settings.

## 1.1 Contributions

We propose the PDA algorithm, which draws inspiration from the dual averaging method originally developed for finite-dimensional convex optimization (Nesterov, 2005, 2009; Xiao, 2009). We iteratively optimize a probability distribution in the form of a Boltzmann distribution, samples from which can be obtained from the Langevin algorithm (see Figure 1). The resulting algorithm has comparable per-iteration cost as gradient descent and can be efficiently implemented.

For optimizing two-layer neural network in the mean-field regime, we establish quantitative global convergence rate of PDA in minimizing an KL-regularized objective: the algorithm requires $\tilde{O}(\epsilon^{-3})$ steps and $\tilde{O}(\epsilon^{-2})$ particles to reach an $\epsilon$-accurate solution, where $\tilde{O}$ hides logarithmic factors. Importantly, our analysis does not couple the learning dynamics with certain continuous time limit, but directly handles the discrete update. This leads to a simpler analysis that covers more general settings. We also derive the generalization bound on the solution obtained by the algorithm. From the viewpoint of the optimization, PDA is an extension of Langevin algorithm to handle entropic-regularized nonlinear functionals on the probability space. Hence we believe our proposed method can also be applied to other distribution optimization problems beyond the training of neural networks.

## 1.2 Related Literature

**Mean field limit of two-layer NNs.** The key observation for the mean field analysis is that when the number of neurons becomes large, the evolution of parameters is well-described by a nonlinear partial differential equation (PDE), which can be viewed as solving an infinite-dimensional *convex* problem (Bengio et al., 2006; Bach, 2017). Global convergence can be derived by studying the limiting PDE (Mei et al., 2018; Chizat and Bach, 2018b; Rotskoff and Vanden-Eijnden, 2018; Sirignano and Spiliopoulos, 2020), yet quantitative convergence rate generally requires additional assumptions.

Javanmard et al. (2019) analyzed a particular RBF network and established linear convergence (up to certain error[1]) for strongly concave target functions. Rotskoff et al. (2019) provided a sublinear rate in continuous time for a modified gradient flow. In the regularized setting, Chizat (2019) obtained local linear convergence under certain non-degeneracy assumption on the objective. Wei et al. (2019) also proved polynomial rate for a perturbed dynamics under weak $\ell_2$ regularization.

---

[1]Note that such error yields sublinear rate with respect to arbitrarily small accuracy $\epsilon$.

Our setting is most related to Hu et al. (2019), who studied the minimization of a nonlinear functional with KL regularization on the probability space, and showed linear convergence (in continuous time) of a particle dynamics named *mean field Langevin dynamics* when the regularization is sufficiently strong. Chen et al. (2020) also considered optimizing a KL-regularized objective in the infinite-width and continuous-time limit, and derived NTK-like convergence guarantee under certain parameter scaling. Compared to these prior works, we directly handle the discrete time update in the mean-field regime, and our analysis covers a wider range of regularization parameters and loss functions.

**Langevin algorithm.** Langevin dynamics can be viewed as optimization in the space of probability measures (Jordan and Kinderlehrer, 1996; Jordan et al., 1998); this perspective has been explored in Wibisono (2018); Durmus et al. (2019). It is known that the continuous-time Langevin diffusion converges exponentially fast to target distributions satisfying certain growth conditions (Roberts and Tweedie, 1996; Mattingly et al., 2002). The discretized *Langevin algorithm* has a sublinear convergence rate that depends on the numerical scheme (Li et al., 2019) and has been studied under various metrics (Dalalyan, 2014; Durmus and Moulines, 2017; Cheng and Bartlett, 2017).

The Langevin algorithm can also optimize certain non-convex objectives (Raginsky et al., 2017; Xu et al., 2018; Erdogdu et al., 2018), in which one finite-dimensional "particle" can attain approximate global convergence due to concentration of Boltzmann distribution around the true minimizer. However, such result often depends on the spectral gap that grows exponentially in dimensionality, which renders the analysis ineffective for neural net optimization in the high-dimensional *parameter space*.

Very recently, convergence of Hamiltonian Monte Carlo in learning certain mean field models has been analyzed in Bou-Rabee and Schuh (2020); Bou-Rabee and Eberle (2021). Compared to these concurrent results, our formulation covers a more general class of potentials, and in the context of two-layer neural network, we provide optimization guarantees for a wider range of loss functions.

## 1.3 Notations

Let $\mathbb{R}_+$ denote the set of non-negative real numbers and $\|\cdot\|_2$ the Euclidean norm. Given a density function $q : \mathbb{R}^p \to \mathbb{R}_+$, we denote the expectation with respect to $q(\theta)\mathrm{d}\theta$ by $\mathbb{E}_q[\cdot]$. For a function $f : \mathbb{R}^p \to \mathbb{R}$, we define $\mathbb{E}_q[f] = \int f(\theta)q(\theta)\mathrm{d}\theta$ when $f$ is integrable. KL is the Kullback-Leibler divergence: $\mathrm{KL}(q\|q') \stackrel{\mathrm{def}}{=} \int q(\theta) \log\left(\frac{q(\theta)}{q'(\theta)}\right) \mathrm{d}\theta$. Let $\mathcal{P}_2$ denote the set of positive densities $q$ on $\mathbb{R}^p$ such that the second-order moment $\mathbb{E}_q[\|\theta\|_2^2] < \infty$ and entropy $-\infty < -\mathbb{E}_q[\log(q)] < +\infty$ are well defined. $\mathcal{N}(0, I_p)$ is the Gaussian distribution on $\mathbb{R}^p$ with mean 0 and covariance matrix $I_p$.

## 2 Problem Setting

We consider the problem of risk minimization with neural networks in the mean field regime. For simplicity, we focus on supervised learning. We here formalize the problem setting and models. Let $\mathcal{X} \subset \mathbb{R}^d$ and $\mathcal{Y} \subset \mathbb{R}$ be the input and output spaces, respectively. For given input data $x \in \mathcal{X}$, we predict a corresponding output $y = h(x) \in \mathcal{Y}$ through a hypothesis function $h : \mathcal{X} \to \mathcal{Y}$.

### 2.1 Neural Network and Mean Field Limit

We adopt a neural network in the mean field regime as a hypothesis function. Let $\Omega = \mathbb{R}^p$ be a parameter space and $h_\theta : \mathcal{X} \to \mathcal{Y}$ ($\theta \in \Omega$) be a bounded function which will be a component of a neural network. We sometimes denote $h(\theta, x) = h_\theta(x)$. Let $q(\theta)\mathrm{d}\theta$ be a probability distribution on the parameter space $\Omega$ and $\Theta = \{\theta_r\}_{r=1}^M$ be the set of parameters $\theta_r$ sampled from $q(\theta)\mathrm{d}\theta$. A hypothesis is defined as an ensemble of $h_{\theta_r}$ as follows:

$$h_\Theta(x) \stackrel{\mathrm{def}}{=} \frac{1}{M} \sum_{r=1}^M h_{\theta_r}(x). \tag{1}$$

A typical example in the literature of the above formulation is a two-layer neural network.

**Example 1** (Two-layer Network). *Let $a_r \in \mathbb{R}$ and $b_r \in \mathbb{R}^d$ ($r \in \{1, 2, \ldots, M\}$) be parameters for output and input layers, respectively. We set $\theta_r = (a_r, b_r)$ and $\Theta = \{\theta_r\}_{r=1}^M$. Denote $h_{\theta_r}(x) \stackrel{\mathrm{def}}{=} \sigma_2(a_r\sigma_1(b_r^\top x))$ ($x \in \mathcal{X}$), where $\sigma_1$ and $\sigma_2$ are smooth activation functions. Then the hypothesis $h_\Theta$ is a two-layer neural network composed of neurons $h_{\theta_r}$: $h_\Theta(x) = \frac{1}{M} \sum_{r=1}^M \sigma_2(a_r\sigma_1(b_r^\top x))$.*

**Remark.** The purpose of $\sigma_2$ in the last layer is to ensure the boundedness of output (e.g., see Assumption 2 in Mei et al. (2018)); this nonlinearity can also be removed if parameters of output layer are fixed. In addition, although we mainly focus on the optimization of two-layer neural network, our proposed method can also be applied to ensemble $h_\Theta$ of deep neural networks $h_{\theta_r}$.

Suppose the parameters $\theta_r$ follow a probability distribution $q(\theta)d\theta$, then $h_\Theta$ can be viewed as a finite-particle discretization of the following expectation,

$$h_q(x) = \mathbb{E}_q[h_\theta(x)]. \tag{2}$$

which we refer to as the *mean field limit* of the neural network $h_\Theta$. As previously discussed, when $h_\Theta$ is overparameterized, optimizing $h_\Theta$ becomes "close" to directly optimizing the probability distribution on the parameter space $\Omega$, for which convergence to the optimal solution may be established under appropriate conditions (Nitanda and Suzuki, 2017; Mei et al., 2018; Chizat and Bach, 2018b). Hence, the study of optimization of $h_q$ with respect to the probability distribution $q(\theta)d\theta$ may shed light on important properties of overparameterized neural networks.

## 2.2 Regularized Empirical Risk Minimization

We briefly outline our setting for regularized expected / empirical risk minimization. The prediction error of a hypothesis is measured by the loss function $\ell(z, y)$ $(z, y \in \mathcal{Y})$, such as the squared loss $\ell(z, y) = 0.5(z - y)^2$ for regression, or the logistic loss $\ell(z, y) = \log(1 + \exp(-yz))$ for binary classification. Let $\mathcal{D}$ be a data distribution over $\mathcal{X} \times \mathcal{Y}$. For expected risk minimization, the distribution $\mathcal{D}$ is set to the true data distribution; whereas for empirical risk minimization, we take $\mathcal{D}$ to be the empirical distribution defined by training data $\{(x_i, y_i)\}_{i=1}^n$ $(x_i \in \mathcal{X}, y_i \in \mathcal{Y})$ independently sampled from the data distribution. We aim to minimize the expected / empirical risk together with a regularization term, which controls the model complexity and also stabilizes the optimization. The regularized objective can be written as follows: for $\lambda_1, \lambda_2 > 0$,

$$\min_{q \in \mathcal{P}_2} \left\{ \mathcal{L}(q) \stackrel{\text{def}}{=} \mathbb{E}_{(X,Y) \sim \mathcal{D}}[\ell(h_q(X), Y)] + R_{\lambda_1, \lambda_2}(q) \right\}, \tag{3}$$

where $R_{\lambda_1, \lambda_2}$ is a regularization term composed of the weighted sum of the second-order moment and negative entropy with regularization parameters $\lambda_1, \lambda_2$:

$$R_{\lambda_1, \lambda_2}(q) \stackrel{\text{def}}{=} \lambda_1 \mathbb{E}_q[\|\theta\|_2^2] + \lambda_2 \mathbb{E}_q[\log(q)]. \tag{4}$$

Note that this regularization is the KL divergence of $q$ from a Gaussian distribution. In our setting, such regularization ensures that the Gibbs distributions $q_*^{(t)}$ specified in Section 3 are well defined.

While our primary focus is the optimization of the objective (3), we can also derive a generalization error bound for the empirical risk minimizer of order of $O(n^{-1/2})$ for both the regression and binary classification settings, following Chen et al. (2020). We defer the details to Appendix D.

## 2.3 The Langevin Algorithm

Before presenting our proposed method, we briefly review the Langevin algorithm. For a given smooth potential function $f : \Omega \to \mathbb{R}$, the Langevin algorithm performs the following update: given the initial $\theta^{(1)} \sim q^{(1)}(\theta)d\theta$, step size $\eta > 0$, and Gaussian noise $\zeta^{(k)} \sim \mathcal{N}(0, I_p)$,

$$\theta^{(k+1)} \leftarrow \theta^{(k)} - \eta \nabla_\theta f(\theta^{(k)}) + \sqrt{2\eta} \zeta^{(k)}. \tag{5}$$

Under appropriate conditions on $f$, it is known that $\theta^{(t)}$ converges to a stationary distribution proportional to $\exp(-f(\cdot))$ in terms of KL divergence at a linear rate (e.g., Vempala and Wibisono (2019)) up to $O(\eta)$-error, where we hide additional factors in the big-$O$ notation.

Alternatively, note that when the normalization constant $\int \exp(-f(\theta))d\theta$ exists, the Boltzmann distribution in proportion to $\exp(-f(\cdot))$ is the solution of the following optimization problem,

$$\min_{q:\text{density}} \left\{ \mathbb{E}_q[f] + \mathbb{E}_q[\log(q)] \right\}. \tag{6}$$

Hence we may interpret the Langevin algorithm as approximately solving an entropic regularized linear functional (i.e., free energy functional) on the probability space. This connection between

sampling and optimization (see Dalalyan (2017); Wibisono (2018); Durmus et al. (2019)) enables us to employ the Langevin algorithm to obtain (samples from) the closed-form Boltzmann distribution which is the minimizer of (6); for example, many Bayesian inference problems fall into this category.

However, the objective (3) that we aim to optimize is beyond the scope of Langevin algorithm – due to the *nonlinearity* of loss $\ell(z, y)$ with respect to $z$, the stationary distribution cannot be described as a closed-form solution of (6). To overcome this limitation, we develop the particle dual averaging (PDA) algorithm which efficiently solves (3) with quantitative runtime guarantees.

# 3 Proposed Method

We now propose the *particle dual averaging* method to approximately solve the problem (3) by optimizing a two-layer neural network in the mean field regime; we also introduce the mean field limit of the proposed method to explain the algorithmic intuition and develop the convergence analysis.

## 3.1 Particle Dual Averaging

Our proposed particle dual averaging method (Algorithm 1) is an optimization algorithm on the space of probability measures. The algorithm consists of an inner loop and outer loop; we run Langevin algorithm in inner loop to approximate a Gibbs distribution, which is optimized in the outer loop so that it converges to the optimal distribution $q_*$. This outer loop update is designed to extend the classical dual averaging scheme (Nesterov, 2005, 2009; Xiao, 2009) to infinite dimensional optimization problems (described in Section 3.2). Below we provide a more detailed explanation.

- In the outer loop, the last iterate $\tilde{\Theta}^{(t)}$ of the previous inner loop is given. We compute $\partial_z \ell(h_{\tilde{\Theta}^{(t)}}(x_t), y_t)$, which is a component of the Gibbs potential[2], and initialize a set of particles $\Theta^{(1)}$ at $\tilde{\Theta}^{(t)}$. In Appendix B we introduce a different "restarting" scheme for the initialization.

- In the inner loop, we run the Langevin algorithm (noisy gradient descent) starting from $\Theta^{(1)}$, where the gradient at the $k$-th inner step is given by $\nabla_\theta \overline{g}^{(t)}(\theta_r^{(k)})$, which is a sum of weighted average of $\partial_z \ell(h_{\tilde{\Theta}^{(s)}}(x_s), y_s) \partial_\theta h(\theta_r^{(k)}, x_s)$ and the gradient of $\ell_2$-regularization (see Algorithm 1).

---

**Algorithm 1** Particle Dual Averaging (PDA)

---

**Input:** data distribution $\mathcal{D}$, initial density $q^{(1)}$, number of outer-iterations $T$, learning rates $\{\eta_t\}_{t=1}^T$, number of inner-iterations $\{T_t\}_{t=1}^T$

Randomly draw i.i.d. initial parameters $\tilde{\theta}_r^{(1)} \sim q^{(1)}(\theta)\mathrm{d}\theta$ ($r \in \{1, 2, \ldots, M\}$)

$\tilde{\Theta}^{(1)} \leftarrow \{\tilde{\theta}_r^{(1)}\}_{r=1}^M$

**for** $t = 1$ **to** $T$ **do**

    Randomly draw data $(x_t, y_t)$ from $\mathcal{D}$

    $\Theta^{(1)} = \{\theta_r^{(1)}\}_{r=1}^M \leftarrow \tilde{\Theta}^{(t)}$

    **for** $k = 1$ **to** $T_t$ **do**

        Run inexact noisy gradient descent for $r \in \{1, 2, \ldots, M\}$

        $\nabla_\theta \overline{g}^{(t)}(\theta_r^{(k)}) \leftarrow \frac{2}{\lambda_2(t+2)(t+1)} \sum_{s=1}^t s \partial_z \ell(h_{\tilde{\Theta}^{(s)}}(x_s), y_s) \partial_\theta h(\theta_r^{(k)}, x_s) + \frac{2\lambda_1 t}{\lambda_2(t+2)} \theta_r^{(k)}$

        $\theta_r^{(k+1)} \leftarrow \theta_r^{(k)} - \eta_t \nabla_\theta \overline{g}^{(t)}(\theta_r^{(k)}) + \sqrt{2\eta_t}\zeta_r^{(k)}$ (i.i.d. Gaussian noise $\zeta_r^{(k)} \sim \mathcal{N}(0, I_p)$)

    **end for**

    $\tilde{\Theta}^{(t+1)} \leftarrow \Theta^{(T_t+1)} = \{\theta_r^{(T_t+1)}\}_{r=1}^M$

**end for**

Randomly pick up $t \in \{2, 3, \ldots, T+1\}$ following the probability $\mathbb{P}[t] = \frac{2t}{T(T+3)}$ and return $h_{\tilde{\Theta}^{(t)}}$

---

Figure 1 provides a pictorial illustration of Algorithm 1. Note that this procedure is a slight modification of the normal gradient descent algorithm: the first term of $\nabla_\theta \overline{g}^{(t)}$ is similar to the gradient of the loss $\partial_{\theta_r} \ell(h_{\Theta^{(k)}}(x), y) \sim \partial_z \ell(h_{\Theta^{(k)}}(x), y) \partial_\theta h(\theta_r^{(k)}, x)$ where $\Theta^{(k)} = \{\theta_r^{(k)}\}_{r=1}^M$. Indeed, if we

---

[2]In Algorithm 1, the terms $\partial_z \ell(h_{\tilde{\Theta}^{(s)}}(x_s), y_s)$ appear in inner loop; but note that these terms only need to be computed in outer loop because they are independent to the inner loop iterates.

set the number of inner-iterations $T_t = 1$ and replace the direction $\nabla_\theta \overline{g}^{(t)}(\theta_r^{(k)})$ with the gradient of the $L_2$-regularized loss, then PDA exactly reduces to the standard noisy gradient descent algorithm considered in Mei et al. (2018). Algorithm 1 can be extended to the minibatch variant in the obvious manner; for efficient implementation in the empirical risk minimization setting see Appendix E. 1.

## 3.2 Mean Field View of PDA

In this subsection we discuss the mean field limit of PDA and explain its algorithmic intuition. Note that the inner loop of Algorithm 1 is the Langevin algorithm with $M$ particles, which optimizes the potential function given by the weighted sum:

$$\overline{g}^{(t)}(\theta) = \frac{2}{\lambda_2(t+2)(t+1)} \sum_{s=1}^{t} s \left( \partial_z \ell(h_{\tilde{\Theta}^{(s)}}(x_s), y_s) h(\theta, x_s) + \lambda_1 \|\theta\|_2^2 \right).$$

Due to the rapid convergence of Langevin algorithm outlined in Subsection 2.3, the particles $\theta_r^{(k+1)}$ ($r \in \{1, \ldots, M\}$) can be regarded as (approximate) samples from the Boltzmann distribution: $\exp\left(-\overline{g}^{(t)}\right)$. Hence, the inner loop of PDA returns an $M$-particle approximation of some stationary distribution, which is then modified in the outer loop. Importantly, the update on the stationary distribution is designed so that the algorithm converges to the optimal solution of the problem (3).

We now introduce the *mean field limit* of PDA, i.e., taking the number of particles $M \to \infty$ and directly optimizing the problem (3) over $q$. We refer to this mean field limit simply as the dual averaging (DA) algorithm. The dual averaging method was originally developed for the convex optimization in finite-dimensional spaces (Nesterov, 2005, 2009; Xiao, 2009), and here we adapt it to optimization on the probability space. The detail of the DA algorithm is described in Algorithm 2.

---

**Algorithm 2** Dual Averaging (DA)

---

**Input:** data distribution $\mathcal{D}$ and initial density $q^{(1)}$

**for** $t = 1$ **to** $T$ **do**

    Randomly draw a data $(x_t, y_t)$ from $\mathcal{D}$

    $g^{(t)} \leftarrow \partial_z \ell(h_{q^{(t)}}(x_t), y_t) h(\cdot, x_t) + \lambda_1 \| \cdot \|_2^2$

    Obtain an approximation $q^{(t+1)}$ of the density function $q_*^{(t+1)} \propto \exp\left(-\frac{\sum_{s=1}^{t} 2s g^{(s)}}{\lambda_2(t+2)(t+1)}\right)$

**end for**

Randomly pick up $t \in \{2, 3, \ldots, T+1\}$ following the probability $\mathbb{P}[t] = \frac{2t}{T(T+3)}$ and return $h_{q^{(t)}}$

---

Algorithm 2 iteratively updates the density function $q_*^{(t+1)} \in \mathcal{P}_2$ which is a solution to the objective:

$$\min_{q \in \mathcal{P}_2} \left\{ \mathbb{E}_q \left[ \sum_{s=1}^{t} s g^{(s)} \right] + \frac{\lambda_2}{2}(t+2)(t+1) \mathbb{E}_q[\log(q)] \right\}, \tag{7}$$

where the function $g^{(t)} = \partial_z \ell(h_{q^{(t)}}(x_t), y_t) h(\cdot, x_t) + \lambda_1 \| \cdot \|_2^2$ is the functional derivative of $\ell(h_q(x_{i_i}), y_t) + \lambda_1 \mathbb{E}_q[\|\theta\|_2^2]$ with respect to $q$ at $q^{(t)}$. In other words, the objective (7) is the sum of weighted average of linear approximations of loss function and the entropic regularization in the space of probability distributions. In this sense, the DA method can be seen as an extension of the Langevin algorithm to handle entropic regularized nonlinear functionals on the probability space by iteratively *linearizing* the objective.

To sum up, we may interpret the DA method as approximating the optimal distribution $q_*$ by iteratively optimizing $q_*^{(t)}$, which takes the form of a Boltzmann distribution. In the inner loop of the PDA algorithm, we obtain $M$ (approximate) samples from $q_*^{(t)}$ via the Langevin algorithm. In other words, PDA can be viewed as a finite-particle approximation of DA – indeed, the stationary distributions obtained in PDA converges to $q_*^{(t+1)}$ by taking $M \to \infty$. In the following section, we present the convergence rate of the DA method, and also take into account the iteration complexity of the Langevin algorithm; we defer the finite-particle approximation error analysis to Appendix C.

# 4 Convergence Analysis

We now provide quantitative global convergence guarantee for our proposed method in discrete time. We first derive the outer loop complexity, assuming approximate optimality of the inner loop iterates, which we then verify in the inner loop analysis. The total complexity is then simply obtained by combining the outer- and inner-loop runtime.

## 4.1 Outer Loop Complexity

We first analyze the convergence rate of the dual averaging (DA) method (Algorithm 2). Our analysis will be made under the following assumptions.

**Assumption 1.**

(A1) $\mathcal{Y} \subset [-1, 1]$. $\ell(z, y)$ is a smooth convex function w.r.t. $z$ and $|\partial_z \ell(z, y)| \leq 2$ for $y, z \in \mathcal{Y}$.

(A2) $|h(\theta, x)| \leq 1$ and $h(\theta, x)$ is smooth with respect to $\theta$ for $x \in \mathcal{X}$.

(A3) $\mathrm{KL}(q^{(t+1)} \| q_*^{(t+1)}) \leq 1/t^2$.

**Remark.** (A2) is satisfied by smooth activation functions such as sigmoid and tanh. Many loss functions including the squared loss and logistic loss satisfy (A1) under the boundedness assumptions $\mathcal{Y} \subset [-1, 1]$ and $|h_\theta(x)| \leq 1$. Note that constants in (A1) and (A2) are defined for simplicity and can be relaxed to any value. (A3) specifies the precision of approximate solutions of sub-problems (7) to guarantee the global convergence of Algorithm 2, which we verify in our inner loop analysis.

We first introduce the following quantity for $q \in \mathcal{P}_2$,

$$e(q) \overset{\mathrm{def}}{=} \mathbb{E}_q[\log(q)] - \frac{4}{\lambda_2} - \frac{p}{2}\left( \exp\left( \frac{4}{\lambda_2} \right) + \log\left( \frac{3\pi\lambda_2}{\lambda_1} \right) \right).$$

Observe that the expression consists of the negative entropy minus its lower bound for $q_*^{(t)}$ under Assumption (A1), (A2); in other words $e(q_*^{(t)}) \geq 0$. We have the following convergence rate of DA[3].

**Theorem 1** (Convergence of DA). *Under Assumptions* (A1), (A2), *and* (A3), *for arbitrary* $q_* \in \mathcal{P}_2$, *iterates of the DA method (Algorithm 2) satisfies*

$$\frac{2}{T(T+3)} \sum_{t=2}^{T+1} t\left( \mathbb{E}[\mathcal{L}(q^{(t)})] - \mathcal{L}(q_*) \right)$$

$$\leq O\Big( \frac{1}{T^2}\left( 1 + \lambda_1 \mathbb{E}_{q_*}\left[ \|\theta\|_2^2 \right] \right) + \frac{\lambda_2 e(q_*)}{T} + \frac{\lambda_2}{T}(1 + \exp(8/\lambda_2))p^2 \log^2(T+2) \Big),$$

*where the expectation* $\mathbb{E}[\mathcal{L}(q^{(t)})]$ *is taken with respect to the history of examples.*

Theorem 1 demonstrates the convergence rate of Algorithm 2 to the optimal value of the regularized objective (3) in expectation. Note that $\frac{2}{T(T+3)} \sum_{t=2}^{T+1} t\mathbb{E}[\mathcal{L}(q^{(t)})]$ is the expectation of $\mathbb{E}[\mathcal{L}(q^{(t)})]$ according to the probability $\mathcal{P}[t] = \frac{2t}{T(T+3)}$ $(t \in \{2, \ldots, T+1\})$ as specified in Algorithm 2. If we take $p, \lambda_1, \lambda_2$ as constants and use $\tilde{O}$ to hide the logarithmic terms, we can deduce that after $\tilde{O}(\epsilon^{-1})$ iterations, an $\epsilon$-accurate solution of the optimal distribution: $\mathcal{L}(q) \leq \inf_{q \in \mathcal{P}_2} \mathcal{L}(q) + \epsilon$ is achieved in expectation. Importantly, this convergence rate applies to *any* choice of regularization parameters, in contrast to the strong regularization required in Hu et al. (2019); Jabir et al. (2019).

On the other hand, due to the exponential dependence on $\lambda_2^{-1}$, our convergence rate is not informative under weak regularization $\lambda_2 \to 0$. Such dependence follows from the classical LSI perturbation lemma (Holley and Strook, 1987), which is likely unavoidable for Langevin-based methods in the most general setting (Menz and Schlichting, 2014), unless additional assumptions are imposed (e.g., a student-teacher setup); we intend to further investigate these conditions in future work.

---

[3]In Appendix B we introduce a more general version of Theorem 1 that allows for inexact $h_{q^{(t)}}(x)$, which simplifies the analysis of finite-particle discretization presented in Appendix C.

## 4.2 Inner Loop Complexity

In order to derive the total complexity (i.e., taking both the outer loop and inner loop into account) towards a required accuracy, we also need to estimate the iteration complexity of Langevin algorithm. We utilize the following convergence result under the log-Sobolev inequality (Definition A):

**Theorem 2** (Vempala and Wibisono (2019)). *Consider a probability density $q(\theta) \propto \exp(-f(\theta))$ satisfying the log-Sobolev inequality with constant $\alpha$, and assume $f$ is smooth and $\nabla f$ is $L$-Lipschitz, i.e., $\|\nabla_\theta f(\theta) - \nabla_\theta f(\theta')\|_2 \le L\|\theta - \theta'\|_2$. If we run the Langevin algorithm (5) with learning rate $0 < \eta \le \frac{\alpha}{4L^2}$ and let $q^{(k)}(\theta)\mathrm{d}\theta$ be a probability distribution that $\theta^{(k)}$ follows, then we have,*

$$\mathrm{KL}(q^{(k)}\|q) \le \exp(-\alpha\eta k)\mathrm{KL}(q^{(1)}\|q) + 8\alpha^{-1}\eta p L^2.$$

Theorem 2 implies that a $\delta$-accurate solution in KL divergence can be obtained by the Langevin algorithm with $\eta \le \frac{\alpha}{4L^2}\min\left\{1, \frac{\delta}{4p}\right\}$ and $\frac{1}{\alpha\eta}\log\frac{2\mathrm{KL}(q^{(1)}\|q)}{\delta}$-iterations.

Since the optimal solution of a sub-problem in DA (Algorithm 2) takes the forms of $q_*^{(t+1)} \propto \exp\left(-\frac{\sum_{s=1}^t 2sg^{(s)}}{\lambda_2(t+2)(t+1)}\right)$, we can verify the LSI and determine the constant for $q_*^{(t+1)}(\theta)\mathrm{d}\theta$ based on the LSI perturbation lemma from Holley and Stroock (1987) (see Lemma B and Example 2 in Appendix A. 2). Consequently, we can apply Theorem 2 to $q_*^{(t+1)}$ for the inner loop complexity when $\nabla_\theta \log q_*^{(t+1)}$ is Lipschitz continuous, which motivates us to introduce the following assumption.

**Assumption 2.**

(A4) $\partial_\theta h(\cdot, x)$ *is 1-Lipschitz continuous*: $\|\partial_\theta h(\theta, x) - \partial_\theta h(\theta', x)\|_2 \le \|\theta - \theta'\|_2$, $\forall x \in \mathcal{X}$, $\theta, \theta' \in \Omega$.

**Remark.** **(A4)** is parallel to (Mei et al., 2018, Assumption A3), and is satisfied by two-layer neural network in Example 1 when the output or input layer is fixed and the input space $\mathcal{X}$ is compact. We remark that this assumption can be relaxed to Hölder continuity of $\partial_\theta h(\cdot, x)$ via the recent result of Erdogdu and Hosseinzadeh (2020), which allows us to extend Theorem 1 to general $L_p$-norm regularizer for $p > 1$. For now we work with **(A4)** for simplicity of the presentation and proof.

Set $\delta_{t+1}$ to be the desired accuracy of an approximate solution $q^{(t+1)}$ specified in **(A3)**: $\delta_{t+1} = 1/(t+1)^2$, we have the following guarantee for the inner loop.

**Corollary 1** (Inner Loop Complexity). *Under* **(A1)**, **(A2)**, *and* **(A4)**, *if we run the Langevin algorithm with step size $\eta_t = O\left(\frac{\lambda_1 \lambda_2 \delta_{t+1}}{p(1+\lambda_1)^2 \exp(8/\lambda_2)}\right)$ on (7), then an approximate solution satisfying $\mathrm{KL}(q^{(t+1)}\|q_*^{(t+1)}) \le \delta_{t+1}$ can be obtained within $O\left(\frac{\lambda_2 \exp(8/\lambda_2)}{\lambda_1 \eta_t}\log\frac{2\mathrm{KL}(q^{(t)}\|q_*^{(t+1)})}{\delta_{t+1}}\right)$-iterations. Moreover, $\mathrm{KL}(q^{(t)}\|q_*^{(t+1)})$ $(t \in \{1, 2, \ldots, T+1\})$ are uniformly bounded with respect to $t$ as long as $q^{(1)}$ is a Gaussian distribution and* **(A3)** *is satisfied.*

We comment that for the inner loop we utilized the *overdamped* Langevin algorithm, since it is the most standard and commonly used sampling method for the objective (7). Our analysis can easily incorporate other inner loop updates such as the underdamped Langevin algorithm (Cheng et al., 2018; Eberle et al., 2019) or the Metropolis-adjusted Langevin algorithm (Roberts and Tweedie, 1996; Dwivedi et al., 2018), which may improve the iteration complexity.

## 4.3 Total Complexity

Combining Theorem 1 and Corollary 1, we can now derive the total complexity of our proposed algorithm. For simplicity, we take $p, \lambda_1, \lambda_2$ as constants and hide logarithmic terms in $\tilde{O}$ and $\tilde{\Theta}$. The following corollary establishes a $\tilde{O}(\epsilon^{-3})$ total iteration complexity to obtain an $\epsilon$-accurate solution in expectation because $T_t = \tilde{\Theta}(t^2) = \tilde{O}(\epsilon^{-2})$ for $t \le T$.

**Corollary 2** (Total Complexity). *Let $\epsilon > 0$ be an arbitrary desired accuracy and $q^{(1)}$ be a Gaussian distribution. Under assumptions* **(A1)**, **(A2)**, **(A3)**, *and* **(A4)**, *if we run Algorithm 2 for $T = \tilde{\Theta}(\epsilon^{-1})$ iterations on the outer loop, and the Langevin algorithm with step size $\eta_t = \Theta\left(\frac{\lambda_1 \lambda_2 \delta_{t+1}}{p(1+\lambda_1)^2 \exp(8/\lambda_2)}\right)$ for $T_t = \tilde{\Theta}(\eta_t^{-1})$ iterations on the inner loop, then an $\epsilon$-accurate solution: $\mathcal{L}(q) \le \inf_{q \in \mathcal{P}_2} \mathcal{L}(q) + \epsilon$ of the objective (3) is achieved in expectation.*

**Quantitative convergence guarantee.** To translate the above convergence rate result to the finite-particle PDA (Algorithm 1), we also characterize the finite-particle discretization error in Appendix C. For the particle complexity analysis, we consider two versions of particle update: (*i*) the *warm-start* scheme described in Algorithm 1, in which $\Theta^{(1)}$ is initialized at the last iterate $\tilde{\Theta}^{(t)}$ of the previous inner loop, and (*ii*) the *resampling* scheme, in which $\Theta^{(1)}$ is initialized from the initial distribution $q^{(1)}(\theta)\mathrm{d}\theta$ (see Appendix B for details). Remarkably, for the resampling scheme, we provide convergence rate guarantee in time- and space-discretized settings that is *polynomial in both the iterations and particle size*; specifically, the particle complexity of $\tilde{O}(\epsilon^{-2})$, together with the total iteration complexity of $\tilde{O}(\epsilon^{-3})$, suffices to obtain an $\epsilon$-accurate solution to the objective (3) (see Appendix B and C for precise statement).

## 5 Experiments

### 5.1 Experiment Setup

We employ our proposed algorithm in both synthetic student-teacher settings (see Figure 2(a)(b)) and real-world dataset (see Figure 2(c)). For the student-teacher setup, the labels are generated as $y_i = f_*(x_i) + \varepsilon_i$, where $f_*$ is the teacher model (target function), and $\varepsilon$ is zero-mean i.i.d. label noise. For the student model $f$, we follow Mei et al. (2018, Section 2.1) and parameterize a two-layer neural network with fixed second layer as:

$$f(x) = \frac{1}{M^\alpha} \sum_{r=1}^{M} \sigma(w_r^\top x + b_r), \tag{8}$$

which we train to minimize the objective (3) using PDA. Note that $\alpha = 1$ corresponds to the mean field regime (which we are interested in), whereas setting $\alpha = 1/2$ leads to the kernel (NTK) regime[4].

**Synthetic student-teacher setting.** For Figure 2(a)(b) we design synthetic experiments for both regression and classification tasks, where the student model is a two-layer tanh network with $M = 500$. For regression, we take the target function $f_*$ to be a multiple-index model with $m$ neurons: $f_*(x) = \frac{1}{\sqrt{m}} \sum_{i=1}^{m} \sigma_*(\langle w_i^*, x \rangle)$, and the input is drawn from a unit Gaussian $\mathcal{N}(0, I_p)$. For binary classification, we consider a simple two-dimensional dataset from `sklearn.datasets.make_circles` (Pedregosa et al., 2011), in which the goal is to separate two groups of data on concentric circles (red and blue in Figure 2(b)). We include additional experimental results in Appendix F.

**PDA hyperparameters.** We optimize the *squared loss* for regression and the *logistic loss* for binary classification. The model is trained by PDA with batch size 50. We scale the number of inner loop steps $T_t$ with $t$, and the step size $\eta_t$ with $1/\sqrt{t}$, where $t$ is the outer loop iteration; this heuristic is consistent with the required inner-loop accuracy in Theorem 1 and Proposition 2.

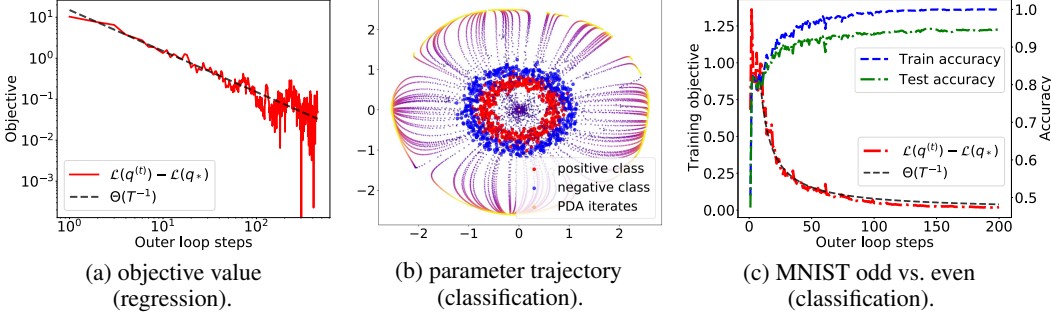

(a) objective value (regression).  (b) parameter trajectory (classification).  (c) MNIST odd vs. even (classification).

Figure 2: (a) Iteration complexity of PDA: the $O(T^{-1})$ rate on the outer loop agrees with Theorem 1. (b) Parameter trajectory of PDA: darker color (purple) indicates earlier in training, and vice versa. (c) odd vs. even classification on MNIST; we report the training loss (red) as well as the train and test accuracy (blue and green).

---

[4]We use the term *kernel regime* only to indicate the parameter scaling $\alpha$; this does not necessarily imply that the NTK linearization is an accurate description of the trained model.

## 5.2 Empirical Findings

**Convergence rate.** In Figure 2(a) we verify the $O(T^{-1})$ iteration complexity of the outer loop in Theorem 1. We apply PDA to optimize the expected risk (analogous to one-pass SGD) in the regression setting, in which the input dimensionality $p = 1$ and the target function is a single-index model ($m = 1$) with tanh activation. We employ the *resampled* update (i.e., without warm-start; see Appendix B) with hyperparameters $\lambda_1 = 10^{-2}, \lambda_2 = 10^{-3}$. To compute the entropy in the objective (3), we adopt the $k$-nearest neighbors estimator (Kozachenko and Leonenko, 1987) with $k = 10$.

**Presence of feature learning.** In Figure 2(b) we visualize the evolution of neural network parameters optimized by PDA in a 2-dimensional classification problem. Due to structure of the input data (concentric rings), we expect that for a two-layer neural network to be a good separator, its parameters should also distribute on a circle. Indeed the converged solution of PDA (bright yellow) agrees with this intuition and demonstrates that PDA learns useful features beyond the kernel regime.

**Binary classification on MNIST.** In Figure 2(c) we report the training and test performance of PDA in separating odd vs. even digits from the MNIST dataset. We subsample $n = 2500$ training examples with binary labels, and learn a two-layer tanh network with width $M = 2500$. We use the resampled update of PDA to optimize the cross entropy loss, with hyperparameters $\lambda_1 = 10^{-2}, \lambda_2 = 10^{-4}$. Observe that the algorithm achieves good generalization performance (green) and roughly maintains[5] the $O(T^{-1})$ iteration complexity (red) in optimizing the training objective (3).

## Conclusion

We proposed the particle dual averaging (PDA) algorithm for optimizing two-layer neural networks in the mean field regime. Leveraging tools from finite-dimensional convex optimization developed in the original dual averaging method, we established *quantitative* convergence rate of PDA for regularized empirical and expected risk minimization. We also provided particle complexity analysis and generalization bounds for both regression and classification problems. Our theoretical findings are aligned with experimental results on neural network optimization. Looking forward, we plan to investigate specific problem instances in which convergence rate can be obtained under vanishing regularization. It is also important to consider accelerated variants of PDA to further improve the convergence rate in the empirical risk minimization setting. Another interesting direction would be to explore other applications of PDA beyond two-layer neural networks, such as deep models (Araújo et al., 2019; Nguyen and Pham, 2020; Lu et al., 2020; Pham and Nguyen, 2021), as well as other optimization problems for entropic regularized nonlinear functional.

## Acknowledgment

The authors would like to thank Murat A. Erdogdu and anonymous NeurIPS reviewers for their helpful feedback. AN was partially supported by JSPS Kakenhi (19K20337) and JST-PRESTO (JPMJPR1928). DW was partially supported by NSERC and LG Electronics. TS was partially supported by JSPS KAKENHI (18H03201), Japan Digital Design and JST CREST.

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
