# Table of Contents

# MISSING PROOFS

## A  Preliminaries

### A. 1  Entropic Regularized Linear Functional

In this section, we explain the property of the optimal solution of the entropic regularized linear functional. We here define the gradient of the negative entropy $\mathbb{E}_q[\log(q)]$ with respect to $q$ over the probability space as $\nabla_q \mathbb{E}_q[\log(q)] = \log(q)$. Note that this gradient is well defined up to constants as a linear operator on the probability space: $q' \mapsto \int (q' - q)(\theta) \log(q(\theta)) \mathrm{d}\theta$. The following lemma shows the strong convexity of the negative entropy.

**Lemma A.** *Let $q, q'$ be probability densities such that the entropy and Kullback-Leibler divergence* $\mathrm{KL}(q'\|q) = \int q'(\theta) \log\left(\frac{q'(\theta)}{q(\theta)}\right) \mathrm{d}\theta$ *are well defined. Then, we have*

$$\mathbb{E}_q[\log(q)] + \int (q' - q)(\theta) \nabla_q \mathbb{E}_q[\log(q)] \mathrm{d}\theta + \mathrm{KL}(q'\|q) = \mathbb{E}_{q'}[\log(q')],$$

$$\mathbb{E}_q[\log(q)] + \int (q' - q)(\theta) \nabla_q \mathbb{E}_q[\log(q)] \mathrm{d}\theta + \frac{1}{2}\|q' - q\|_{L_1(\mathrm{d}\theta)}^2 \leq \mathbb{E}_{q'}[\log(q')].$$

The first equality of this lemma can be shown by the direct computation of the entropy, and the second inequality can be obtained by Pinsker's inequality $\frac{1}{2}\|q' - q\|_{L_1(\mathrm{d}\theta)}^2 \leq \mathrm{KL}(q'\|q)$.

Recall that $\mathcal{P}_2$ is the set of positive densities on $\mathbb{R}^p$ such that the second moment $\mathbb{E}_q[\|\theta\|_2^2] < \infty$ and entropy $-\infty < -\mathbb{E}_q[\log(q)] < +\infty$ are well defined. We here consider the minimization problem of entropic regularized linear functional on $\mathcal{P}_2$. Let $\lambda_1, \lambda_2 > 0$ be positive real numbers and $H : \mathbb{R}^p \to \mathbb{R}$ be a bounded continuous function.

$$\min_{q \in \mathcal{P}_2} \left\{ F(q) \stackrel{\text{def}}{=} \mathbb{E}_q[H(\theta)] + \lambda_1 \mathbb{E}_q[\|\theta\|_2^2] + \lambda_2 \mathbb{E}_q[\log(q(\theta))] \right\}. \tag{9}$$

Then, we can show $q \propto \exp\left(-\frac{H(\theta) + \lambda_1 \|\theta\|_2^2}{\lambda_2}\right)$ is an optimal solution of the problem (9) as follow. Clearly, $q \in \mathcal{P}_2$ and the assumption on $q$ in Lemma A with $q' \in \mathcal{P}_2$ holds. Hence, for $\forall q' \in \mathcal{P}_2$,

$$F(q) = \mathbb{E}_q[H(\theta)] + \lambda_1 \mathbb{E}_q[\|\theta\|_2^2] + \lambda_2 \mathbb{E}_q[\log(q(\theta))]$$

$$= \mathbb{E}_{q'}[H(\theta)] + \lambda_1 \mathbb{E}_{q'}[\|\theta\|_2^2] + \lambda_2 \mathbb{E}_{q'}[\log(q'(\theta))]$$

$$+ \int (q - q')(\theta) \left(H(\theta) + \lambda_1 \|\theta\|_2^2\right) \mathrm{d}\theta + \lambda_2 \left(\mathbb{E}_q[\log(q(\theta))] - \mathbb{E}_{q'}[\log(q'(\theta))]\right)$$

$$= F(q') + \int (q - q')(\theta) \left(H(\theta) + \lambda_1 \|\theta\|_2^2\right) \mathrm{d}\theta + \lambda_2 \left(\mathbb{E}_q[\log(q(\theta))] - \mathbb{E}_{q'}[\log(q'(\theta))]\right)$$

$$\leq F(q') + \int (q - q')(\theta) \left(H(\theta) + \lambda_1 \|\theta\|_2^2\right) \mathrm{d}\theta - \lambda_2 \left(\int (q' - q)(\theta) \nabla_q \mathbb{E}_q[\log(q)] \mathrm{d}\theta + \frac{1}{2}\|q' - q\|_{L_1(\mathrm{d}\theta)}^2\right)$$

$$= F(q') + \int (q - q')(\theta) \left(H(\theta) + \lambda_1 \|\theta\|_2^2 + \lambda_2 \log(q(\theta))\right) \mathrm{d}\theta - \frac{\lambda_2}{2}\|q' - q\|_{L_1(\mathrm{d}\theta)}^2$$

$$= F(q') - \frac{\lambda_2}{2}\|q' - q\|_{L_1(\mathrm{d}\theta)}^2. \tag{10}$$

For the inequality we used Lemma A and for the last equality we used $q \propto \exp\left(-\frac{H(\theta) + \lambda_1 \|\theta\|_2^2}{\lambda_2}\right)$. Therefore, we conclude that $q$ is a minimizer of $F$ on $\mathcal{P}_2$ and the strong convexity of $F$ holds at $q$ with respect to $L_1(\mathrm{d}\theta)$-norm. This crucial property is used in the proof of Theorem 1.

### A. 2  Log-Sobolev and Talagrand's Inequalities

The log-Sobolev inequality is useful in establishing the convergence rate of Langevin algorithm.

**Definition A** (Log-Sobolev inequality). *Let $\mathrm{d}\mu = p(\theta)\mathrm{d}\theta$ be a probability distribution with a positive smooth density $p > 0$ on $\mathbb{R}^p$. We say that $\mu$ satisfies the log-Sobolev inequality with constant $\alpha > 0$ if for any smooth function $f : \mathbb{R}^p \to \mathbb{R}$,*

$$\mathbb{E}_\mu[f^2 \log f^2] - \mathbb{E}_\mu[f^2] \log \mathbb{E}_\mu[f^2] \leq \frac{2}{\alpha} \mathbb{E}_\mu[\|\nabla f\|_2^2].$$

This inequality is analogous to strong convexity in optimization: let $d\nu = q(\theta)d\mu$ be a probability distribution on $\mathbb{R}^p$ such that $q$ is smooth and positive. Then, if $\mu$ satisfies the log-Sobolev inequality with $\alpha$, it follows that

$$\mathrm{KL}(\nu\|\mu) \leq \frac{1}{2\alpha}\mathbb{E}_\nu[\|\nabla_\theta \log q\|_2^2].$$

The above relation is directly obtained by setting $f = \sqrt{q}$ in the definition of log-Sobolev inequality. Note that the right hand side is nothing else but the squared norm of functional gradient of $\mathrm{KL}(\nu\|\mu)$ with respect to a transport map for $\nu$.

It is well-known that strong log-concave densities satisfy the LSI with a dimension-free constant (up to the spectral norm of the covariance).

**Example 2** (Bakry and Émery (1985)). *Let $q \propto \exp(-f)$ be a probability density, where $f : \mathbb{R}^p \to \mathbb{R}$ is a smooth function. If there exists $c > 0$ such that $\nabla^2 f \succeq cI_p$, then $q(\theta)d\theta$ satisfies Log-Sobolev inequality with constant $c$.*

In addition, the LSI is preserved under bounded perturbation, as originally shown in Holley and Stroock (1987). We also provide a proof for completeness.

**Lemma B** (Holley and Stroock (1987)). *Let $q(\theta)d\theta$ be a probability distribution on $\mathbb{R}^p$ satisfying the log-Sobolev inequality with a constant $\alpha$. For a bounded function $B : \mathbb{R}^p \to \mathbb{R}$, we define a probability distribution $q_B(\theta)d\theta$ as follows:*

$$q_B(\theta)d\theta = \frac{\exp(B(\theta))q(\theta)}{\mathbb{E}_q[\exp(B(\theta))]}d\theta.$$

*Then, $q_B d\theta$ satisfies the log-Sobolev inequality with a constant $\alpha/\exp(4\|B\|_\infty)$.*

*Proof.* Taking an expectation $\mathbb{E}_{q_B}$ of the Bregman divergence defined by a convex function $x\log x$, for $\forall a > 0$,

$$0 \leq \mathbb{E}_{q_B}\left[f^2(\theta)\log(f^2(\theta)) - (a\log(a) + (\log(a)+1)(f^2(\theta)-a))\right]$$
$$= \mathbb{E}_{q_B}\left[f^2(\theta)\log(f^2(\theta)) - (f^2(\theta)\log(a) + f^2(\theta) - a)\right].$$

Since the minimum is attained at $a = \mathbb{E}_{q_B}[f^2(\theta)]$,

$$0 \leq \mathbb{E}_{q_B}\left[f^2(\theta)\log(f^2(\theta))\right] - \mathbb{E}_{q_B}[f^2(\theta)]\log\mathbb{E}_{q_B}[f^2(\theta)]$$
$$= \inf_{a>0}\mathbb{E}_{q_B}\left[f^2(\theta)\log(f^2(\theta)) - (f^2(\theta)\log(a) + f^2(\theta) - a)\right]$$
$$\leq \exp(2\|B\|_\infty)\inf_{a>0}\mathbb{E}_q\left[f^2(\theta)\log(f^2(\theta)) - (f^2(\theta)\log(a) + f^2(\theta) - a)\right]$$
$$= \exp(2\|B\|_\infty)\left(\mathbb{E}_q\left[f^2(\theta)\log(f^2(\theta))\right] - \mathbb{E}_q[f^2(\theta)]\log\mathbb{E}_q[f^2(\theta)]\right)$$
$$\leq \frac{2\exp(2\|B\|_\infty)}{\alpha}\mathbb{E}_q\left[\|\nabla f\|_2^2\right]$$
$$= \frac{2\exp(2\|B\|_\infty)}{\alpha}\mathbb{E}_{q_B}\left[\frac{\mathbb{E}_q[\exp(B(\theta))]}{\exp(B(\theta))}\|\nabla f\|_2^2\right]$$
$$\leq \frac{2\exp(4\|B\|_\infty)}{\alpha}\mathbb{E}_{q_B}\left[\|\nabla f\|_2^2\right],$$

where we used the non-negativity of the integrand for the second inequality. $\square$

We next introduce Talagrand's inequality.

**Definition B** (Talagrand's inequality). *We say that a probability distribution $q(\theta)d\theta$ satisfies Talagrand's inequality with a constant $\alpha > 0$ if for any probability distribution $q'(\theta)d\theta$,*

$$\frac{\alpha}{2}W_2^2(q', q) \leq \mathrm{KL}(q'\|q),$$

*where $W_2(q', q)$ denotes the 2-Wasserstein distance between $q(\theta)d\theta$ and $q'(\theta)d\theta$.*

The next theorem gives a relationship between KL divergence and 2-Wasserstein distance.

**Theorem C** (Otto and Villani (2000)). *If a probability distribution $q(\theta)d\theta$ satisfies the log-Sobolev inequality with constant $\alpha > 0$, then $q(\theta)d\theta$ satisfies Talagrand's inequality with the same constant.*

# B Proof of Main Results

## B. 1 Extension of Algorithm

In this section, we prove the main theorem that provides the convergence rate of the dual averaging method. We first introduce a slight extension of PDA (Algorithm 1) which incorporates two different initializations at each outer loop step. We refer to the two versions as the *warm-start* and the *resampled* update, respectively. Note that Algorithm 1 in the main text only includes the warm-start update. In Appendix C we provide particle complexity analysis for both updates. We remark that the benefit of resampling strategy is the simplicity of estimation of approximation error $|h_x^{(t)} - h_{q^{(t)}}(x_t)|$, because $h_x^{(t)}$ is composed of i.i.d particles and a simple concentration inequality can be applied to estimate this error.

---

**Algorithm 3** Particle Dual Averaging (*general version*)

---

**Input:** data distribution $\mathcal{D}$, initial density $q^{(1)}$, number of outer-iterations $T$, learning rates $\{\eta_t\}_{t=1}^T$, number of inner-iterations $\{T_t\}_{t=1}^T$

Randomly draw i.i.d. initial parameters $\tilde{\theta}_r^{(1)} \sim q^{(1)}(\theta)\mathrm{d}\theta$ $(r \in \{1, 2, \ldots, M\})$

$\tilde{\Theta}^{(1)} \leftarrow \{\tilde{\theta}_r^{(1)}\}_{r=1}^M$

**for** $t = 1$ **to** $T$ **do**

    Randomly draw a data $(x_t, y_t)$ from $\mathcal{D}$

    **Either** $\Theta^{(1)} = \{\theta_r^{(1)}\}_{r=1}^M \leftarrow \tilde{\Theta}^{(t)}$ (*warm-start*)

    **Or** randomly initialize $\Theta^{(1)}$ from $q^{(1)}(\theta)\mathrm{d}\theta$ (*resampling*)

    **for** $k = 1$ **to** $T_t$ **do**

        Run an inexact noisy gradient descent for $r \in \{1, 2, \ldots, M\}$

        $\nabla_\theta \overline{g}^{(t)}(\theta_r^{(k)}) \leftarrow \frac{2}{\lambda_2(t+2)(t+1)} \sum_{s=1}^t s\partial_z \ell(h_{\tilde{\Theta}^{(s)}}(x_s), y_s)\partial_\theta h(\theta_r^{(k)}, x_s) + \frac{2\lambda_1 t}{\lambda_2(t+2)}\theta_r^{(k)}$

        $\theta_r^{(k+1)} \leftarrow \theta_r^{(k)} - \eta_t \nabla_\theta \overline{g}^{(t)}(\theta_r^{(k)}) + \sqrt{2\eta_t}\zeta_r^{(k)}$ (i.i.d. Gaussian noise $\zeta_r^{(k)} \sim \mathcal{N}(0, I_p)$)

    **end for**

    $\tilde{\Theta}^{(t+1)} \leftarrow \Theta^{(T_t+1)} = \{\theta_r^{(T_t+1)}\}_{r=1}^M$

**end for**

Randomly pick up $t \in \{2, 3, \ldots, T+1\}$ following the probability $\mathbb{P}[t] = \frac{2t}{T(T+3)}$ and return $h_{\tilde{\Theta}^{(t)}}$

---

We also extend the mean field limit (Algorithm 2) to take into account the inexactness in computing $h_{q^{(t)}}(t)$. This relaxation is useful in convergence analysis of Algorithm 3 with resampling because it allows us to regard this method as an instance of the generalized DA method (Algorithm 4) by setting an inexact estimate $h_x^{(t)} = h_{\tilde{\Theta}^{(t)}}(x_t)$, instead of the exact value of $h_{q^{(t)}}(t)$, which is actually used to defined the potential for which Langevin algorithm run in Algorithm 3. This means convergence analysis of Algorithm 4 (Theorem D) immediately provides a convergence guarantee for Algorithm 3 if the discretization error $|h_x^{(t)} - h_{q^{(t)}}(x_t)|$ can be estimated (as in the resampling scheme).

On the other hands, the convergence analysis of warm-start scheme requires the convergence of mean field limit due to certain technical difficulties, that is, we show the convergence of Algorithm 3 with warm-start by coupling the update with its mean field limit (Algorithm 2) and taking into account the discretization error which stems from finite-particle approximation.

We now present generalized version of the outer loop convergence rate of DA. We highlight the tolerance factor $\epsilon$ in the generalized assumption **(A3')** in blue.

**Assumption C.** *Let $\epsilon > 0$ be a given accuracy.*

  **(A1')** $\mathcal{Y} \subset [-1, 1]$. $\ell(z, y)$ *is a smooth convex function w.r.t. $z$ and $|\partial_z \ell(z, y)| \leq 2$ for $y, z \in \mathcal{Y}$ and $\partial\ell(\cdot, y)$ is 1-Lipschitz continuous for $y \in \mathcal{Y}$.*

  **(A2')** $|h_\theta(x)| \leq 1$ *and $h(\theta, x)$ is smooth w.r.t. $\theta$ for $x \in \mathcal{X}$.*

  **(A3')** $\mathrm{KL}(q^{(t+1)} \| q_*^{(t+1)}) \leq 1/t^2$ *and $|h_x^{(t)} - h_{q^{(t)}}(x_t)| \leq \epsilon$ for $t \geq 1$.*

---

**Algorithm 4** Dual Averaging (*general version*)

---

**Input:** data distribution $\mathcal{D}$ and initial density $q^{(1)}$
**for** $t = 1$ **to** $T$ **do**
    Randomly draw a data $(x_t, y_t)$ from $\mathcal{D}$
    Compute an approximation $h_x^{(t)}$ of $h_{q^{(t)}}(x_t)$
    $g^{(t)} \leftarrow \partial_z \ell(h_x^{(t)}, y_t) h(\cdot, x_t) + \lambda_1 \|\cdot\|_2^2$
    Obtain an approximation $q^{(t+1)}$ of the density function $q_*^{(t+1)} \propto \exp\left(-\frac{\sum_{s=1}^t 2sg^{(s)}}{\lambda_2(t+2)(t+1)}\right)$
**end for**
Randomly pick up $t \in \{2, 3, \ldots, T+1\}$ following the probability $\mathbb{P}[t] = \frac{2t}{T(T+3)}$ and return $h_{q^{(t)}}$

---

**Remark.** The new condition of **(A3')** allows for inexactness of computing $h_{q^{(t)}}(x_t)$. When showing solely the convergence of the Algorithm 2 which is the exact mean-field limit, the original assumptions **(A1)**, **(A2)**, and **(A3)** are sufficient, in other words, we can take $\epsilon = 0$ and Lipschitz continuity of $\partial_z \ell(\cdot, y)$ in **(A1')** can be relaxed.

**Theorem D** (Convergence of general DA). *Under Assumptions **(A1')**, **(A2')**, and **(A3')** with $\epsilon \geq 0$, for arbitrary $q_* \in \mathcal{P}_2$, iterates of the general DA method (Algorithm 4) satisfies*

$$\frac{2}{T(T+3)} \sum_{t=2}^{T+1} t\left(\mathbb{E}[\mathcal{L}(q^{(t)})] - \mathcal{L}(q_*)\right)$$

$$\leq 2\epsilon + O\left(\frac{1}{T^2}\left(1 + \lambda_1 \mathbb{E}_{q_*}\left[\|\theta\|_2^2\right]\right) + \frac{\lambda_2 e(q_*)}{T} + \frac{\lambda_2}{T}(1 + \exp(8/\lambda_2))p^2 \log^2(T+2)\right),$$

*where the expectation $\mathbb{E}[\mathcal{L}(q^{(t)})]$ is taken with respect to the history of examples.*

**Notation.** In the proofs, we use the following notations which are consistent with the description of Algorithm 3 and 4:

$$g^{(t)} = \partial_z \ell(h_x^{(t)}, y_t) h(\cdot, x_t) + \lambda_1 \|\cdot\|_2^2,$$

$$\overline{g}^{(t)} = \frac{2}{\lambda_2(t+2)(t+1)} \sum_{s=1}^t sg^{(s)}$$

$$= \frac{2}{\lambda_2(t+2)(t+1)} \sum_{s=1}^t s\partial_z \ell(h_x^{(s)}, y_s) h(\cdot, x_s) + \frac{\lambda_1 t}{\lambda_2(t+2)} \|\cdot\|_2^2,$$

$$q_*^{(t+1)} \propto \exp\left(-\overline{g}^{(t)}\right)$$

$$= \exp\left(-\frac{\sum_{s=1}^t 2sg^{(s)}}{\lambda_2(t+2)(t+1)}\right).$$

When considering the resampling scheme, $h_x^{(t)}$ is set to the approximation $h_{\tilde{\Theta}^{(t)}}(x_t)$, whereas when considering the warm-start scheme, $h_x^{(t)}$ is set to $h_{q^{(t)}}(x_t)$ with the mean field limit $M \to \infty$ and without tolerance ($\epsilon = 0$).

## B. 2 Auxiliary Lemmas

We introduce several auxiliary results used in the proof of Theorem 1 (Theorem D) and Corollary 1. The following lemma provides a tail bound for Chi-squared variables (Laurent and Massart, 2000).

**Lemma C** (Tail bound for Chi-squared variable). *Let $\theta \sim \mathcal{N}(0, \sigma^2 I_p)$ be a Gaussian random variable on $\mathbb{R}^p$. Then, we get for $\forall c \geq p\sigma^2$,*

$$\mathbb{P}\left[\|\theta\|_2^2 \geq 2c\right] \leq \exp\left(-\frac{c}{10\sigma^2}\right).$$

Based on Lemma C, we get the following bound.

**Lemma D.** *Let $\theta \sim \mathcal{N}(0, \sigma^2 I_p)$ be Gaussian random variable on $\Theta = \mathbb{R}^p$. Then, we get for $\forall R \geq p\sigma^2$,*

$$\mathbb{E}\left[\|\theta\|_2^2 \mathbb{1}[\|\theta\|_2^2 > 2R]\right] = \frac{1}{Z}\int_{\|\theta\|_2^2 > 2R} \|\theta\|_2^2 \exp\left(-\frac{\|\theta\|_2^2}{2\sigma^2}\right) d\theta \leq 2(R + 10\sigma^2)\exp\left(-\frac{R}{10\sigma^2}\right),$$

*where $Z = \int \exp\left(-\frac{\|\theta\|_2^2}{2\sigma^2}\right) d\theta$.*

*Proof.* We set $p(\theta) = \exp(-\|\theta\|_2^2/2\sigma^2)/Z$. Then,

$$\int_{\|\theta\|_2^2 > 2R} \|\theta\|_2^2 p(\theta) d\theta = \int_\Theta p(\theta)\mathbb{1}[\|\theta\|_2^2 > 2R]\int_0^\infty \mathbb{1}[\|\theta\|_2^2 > r] dr d\theta$$

$$= \int_\Theta \int_0^\infty p(\theta)\mathbb{1}\left[\|\theta\|_2^2 > \max\{2R, r\}\right] dr d\theta$$

$$\leq 2R \int_\Theta p(\theta)\mathbb{1}\left[\|\theta\|_2^2 > 2R\right] d\theta + \int_\Theta \int_{2R}^\infty p(\theta)\mathbb{1}\left[\|\theta\|_2^2 > r\right] dr d\theta$$

$$= 2R\mathbb{P}[\|\theta\|_2^2 > 2R] + \int_{2R}^\infty \mathbb{P}[\|\theta\|_2^2 > r] dr$$

$$\leq 2R \exp\left(-\frac{R}{10\sigma^2}\right) + \int_{2R}^\infty \exp\left(-\frac{r}{20\sigma^2}\right) dr$$

$$\leq 2(R + 10\sigma^2)\exp\left(-\frac{R}{10\sigma^2}\right).$$

$\square$

**Proposition A** (Continuity). *Let $q_*(\theta) \propto \exp\left(-H(\theta) - \lambda\|\theta\|_2^2\right)$ $(\lambda > 0)$ be a density on $\mathbb{R}^p$ such that $\|H\|_\infty \leq c$. Then, for $\forall \delta > 0$ and a density $\forall q \in \mathcal{P}_2$,*

$$\left|\int \|\theta\|_2^2(q - q_*)(\theta) d\theta\right| \leq \frac{(2 + \delta + 1/\delta)\exp(4c)}{\lambda}\mathrm{KL}(q\|q_*) + \frac{\delta(1 + \delta)p\exp(2c)}{2\lambda},$$

$$\left|\int q(\theta)\log(q(\theta)) d\theta - \int q_*(\theta)\log(q_*(\theta)) d\theta\right| \leq (1 + (2 + \delta + 1/\delta)\exp(4c))\,\mathrm{KL}(q\|q_*) + c\sqrt{2\mathrm{KL}(q\|q_*)}$$

$$+ \frac{\delta(1 + \delta)p\exp(2c)}{2}.$$

*Proof.* Let $\gamma$ be an optimal coupling between $q d\theta$ and $q_* d\theta$. Using Young's inequality, we have

$$\int \|\theta\|_2^2 q(\theta) d\theta = \int \|\theta\|_2^2 d\gamma(\theta, \theta')$$

$$= \int \left(\|\theta - \theta'\|_2^2 + \|\theta'\|_2^2 + 2(\theta - \theta')^\top \theta'\right) d\gamma(\theta, \theta')$$

$$\leq \int \left(\|\theta - \theta'\|_2^2 + \|\theta'\|_2^2 + \frac{1}{\delta}\|\theta - \theta'\|_2^2 + \delta\|\theta'\|_2^2\right) d\gamma(\theta, \theta')$$

$$= (1 + 1/\delta)\int \|\theta - \theta'\|_2^2 d\gamma(\theta, \theta') + (1 + \delta)\int \|\theta'\|_2^2 q_*(\theta') d\theta'$$

$$= (1 + 1/\delta)W_2^2(q, q_*) + (1 + \delta)\int \|\theta'\|_2^2 q_*(\theta') d\theta'. \tag{11}$$

The last term can be bounded as follows:

$$\int \|\theta\|_2^2 q_*(\theta) d\theta = \int \|\theta\|_2^2 \frac{\exp\left(-H(\theta) - \lambda\|\theta\|_2^2\right)}{\int \exp\left(-H(\theta) - \lambda\|\theta\|_2^2\right) d\theta} d\theta$$

$$\leq \exp(2c)\int \|\theta\|_2^2 \frac{\exp\left(-\lambda\|\theta\|_2^2\right)}{\int \exp\left(-\lambda\|\theta\|_2^2\right) d\theta} d\theta$$

$$= \frac{p\exp(2c)}{2\lambda}, \tag{12}$$

where the last equality comes from the variance of Gaussian distribution.

From (11) and (12),

$$\int \|\theta\|_2^2 (q - q_*)(\theta)\mathrm{d}\theta \le (1 + 1/\delta)W_2^2(q, q_*) + \delta \int \|\theta\|_2^2 q_*(\theta)\mathrm{d}\theta$$

$$\le (1 + 1/\delta)W_2^2(q, q_*) + \frac{\delta p \exp(2c)}{2\lambda}.$$

From the symmetry of (11), and applying (11) again with (12),

$$\int \|\theta\|_2^2 (q_* - q)(\theta)\mathrm{d}\theta \le (1 + 1/\delta)W_2^2(q, q_*) + \delta \int \|\theta\|_2^2 q(\theta)\mathrm{d}\theta$$

$$\le (2 + \delta + 1/\delta)W_2^2(q, q_*) + \delta(1 + \delta) \int \|\theta\|_2^2 q_*(\theta)\mathrm{d}\theta$$

$$\le (2 + \delta + 1/\delta)W_2^2(q, q_*) + \frac{\delta(1 + \delta)p \exp(2c)}{2\lambda}.$$

From Lemma B and Example 2, we see $q_*$ satisfies the log-Sobolev inequality with a constant $2\lambda/\exp(4c)$. As a result, $q_*$ satisfies Talagrand's inequality with the same constant from Theorem C. Hence, by combining the above two inequalities, we have

$$\left| \int \|\theta\|_2^2 (q - q_*)(\theta)\mathrm{d}\theta \right| \le (2 + \delta + 1/\delta)W_2^2(q, q_*) + \frac{\delta(1 + \delta)p \exp(2c)}{2\lambda}$$

$$\le \frac{(2 + \delta + 1/\delta)\exp(4c)}{\lambda}\mathrm{KL}(q\|q_*) + \frac{\delta(1 + \delta)p \exp(2c)}{2\lambda}$$

Therefore, we know that

$$\left| \int q(\theta)\log(q(\theta))\mathrm{d}\theta - \int q_*(\theta)\log(q_*(\theta))\mathrm{d}\theta \right|$$

$$\le \mathrm{KL}(q\|q_*) + \left| \int (q_* - q)(\theta)\left( H(\theta) + \lambda\|\theta\|_2^2 \right)\mathrm{d}\theta \right|$$

$$\le \mathrm{KL}(q\|q_*) + c\|q - q_*\|_{L_1(\mathrm{d}\theta)} + (2 + \delta + 1/\delta)\exp(4c)\mathrm{KL}(q\|q_*) + \frac{\delta(1 + \delta)p \exp(2c)}{2}$$

$$\le \mathrm{KL}(q\|q_*) + c\sqrt{2\mathrm{KL}(q\|q_*)} + (2 + \delta + 1/\delta)\exp(4c)\mathrm{KL}(q\|q_*) + \frac{\delta(1 + \delta)p \exp(2c)}{2}.$$

where we used Pinsker's theorem for the last inequality. This finishes the proof. $\square$

**Proposition B** (Maximum Entropy). *Let* $q_*(\theta) \propto \exp\left(-H(\theta) - \lambda\|\theta\|_2^2\right)$ $(\lambda > 0)$ *on* $\mathbb{R}^p$ *be a density such that* $\|H\|_\infty \le c$. *Then,*

$$-\mathbb{E}_{q_*}[\log(q_*)] \le 2c + \frac{p}{2}\left( \exp(2c) + \log\left( \frac{\pi}{\lambda} \right) \right).$$

*Proof.* It follows that

$$-\mathbb{E}_{q_*}[\log(q_*)] = \mathbb{E}_{q_*}[H(\theta) + \lambda\|\theta\|_2^2] + \log \int \exp(-H(\theta) - \lambda\|\theta\|_2^2)\mathrm{d}\theta$$

$$\le c + \lambda\mathbb{E}_{q_*}[\|\theta\|_2^2] + \log \int \exp(c - \lambda\|\theta\|_2^2)\mathrm{d}\theta$$

$$= 2c + \lambda\mathbb{E}_{q_*}[\|\theta\|_2^2] + \log \int \exp(-\lambda\|\theta\|_2^2)\mathrm{d}\theta$$

$$\le 2c + \frac{p \exp(2c)}{2} + \frac{p}{2}\log\left( \frac{\pi}{\lambda} \right),$$

where we used (12) and Gaussian integral for the last inequality. $\square$

**Proposition C** (Boundedness of KL-divergence)**.** *Let $q_*(\theta) \propto \exp\left(-H_*(\theta) - \lambda_* \|\theta\|_2^2\right)$ $(\lambda_* > 0)$ be a density on $\mathbb{R}^p$ such that $\|H_*\|_\infty \leq c_*$, and $q_\sharp(\theta) \propto \exp\left(-H_\sharp(\theta) - \lambda_\sharp \|\theta\|_2^2\right)$ $(\lambda_\sharp > 0)$ be a density on $\mathbb{R}^p$ such that $\|H_\sharp\|_\infty \leq c_\sharp$. Then, for any density q,*

$$\mathrm{KL}(q\|q_*) \leq 4c_* + 2c_\sharp + \frac{3}{2}\left(1 + \frac{\lambda_*}{\lambda_\sharp}\right) p \exp(2c_\sharp) + \frac{p}{2}\log\left(\frac{\lambda_\sharp}{\lambda_*}\right)$$
$$+ \left(1 + 4\left(1 + \frac{\lambda_*}{\lambda_\sharp}\right)\exp(4c_\sharp)\right)\mathrm{KL}(q\|q_\sharp) + c_\sharp\sqrt{2\mathrm{KL}(q\|q_\sharp)}.$$

*Proof.* Applying Proposition A with $\delta = 1$,

$$\mathrm{KL}(q\|q_*) = \int q(\theta)\log\left(\frac{q(\theta)}{q_*(\theta)}\right)\mathrm{d}\theta$$
$$= \int q_\sharp(\theta)\log\left(\frac{q_\sharp(\theta)}{q_*(\theta)}\right)\mathrm{d}\theta + \int (q_\sharp(\theta) - q(\theta))\log(q_*(\theta))\mathrm{d}\theta$$
$$+ \int q(\theta)\log(q(\theta))\mathrm{d}\theta - \int q_\sharp(\theta)\log(q_\sharp(\theta))\mathrm{d}\theta$$
$$\leq \int q_\sharp(\theta)\log\left(\frac{q_\sharp(\theta)}{q_*(\theta)}\right)\mathrm{d}\theta + \int (q(\theta) - q_\sharp(\theta))(H_*(\theta) + \lambda_*\|\theta\|_2^2)\mathrm{d}\theta$$
$$+ (1 + 4\exp(4c_\sharp))\mathrm{KL}(q\|q_\sharp) + c_\sharp\sqrt{2\mathrm{KL}(q\|q_\sharp)} + p\exp(2c_\sharp)$$
$$\leq \int q_\sharp(\theta)\log\left(\frac{q_\sharp(\theta)}{q_*(\theta)}\right)\mathrm{d}\theta + 2c_* + \frac{4\lambda_*\exp(4c_\sharp)}{\lambda_\sharp}\mathrm{KL}(q\|q_\sharp) + \frac{p\lambda_*\exp(2c_\sharp)}{\lambda_\sharp}$$
$$+ (1 + 4\exp(4c_\sharp))\mathrm{KL}(q\|q_\sharp) + c_\sharp\sqrt{2\mathrm{KL}(q\|q_\sharp)} + p\exp(2c_\sharp).$$

We next bound the first term in the last equation as follows.

$$\int q_\sharp(\theta)\log\left(\frac{q_\sharp(\theta)}{q_*(\theta)}\right)\mathrm{d}\theta = \int q_\sharp(\theta)\log\left(\frac{\exp(-H_\sharp(\theta) - \lambda_\sharp\|\theta\|_2^2)}{\exp(-H_*(\theta) - \lambda_*\|\theta\|_2^2)}\right)\mathrm{d}\theta + \log\frac{\int\exp(-H_*(\theta) - \lambda_*\|\theta\|_2^2)\mathrm{d}\theta}{\int\exp(-H_\sharp(\theta) - \lambda_\sharp\|\theta\|_2^2)\mathrm{d}\theta}$$
$$= \int q_\sharp(\theta)\left(H_*(\theta) - H_\sharp(\theta) + (\lambda_* - \lambda_\sharp)\|\theta\|_2^2\right)\mathrm{d}\theta$$
$$+ \log\int\exp(-H_*(\theta) - \lambda_*\|\theta\|_2^2)\mathrm{d}\theta - \log\int\exp(-H_\sharp(\theta) - \lambda_\sharp\|\theta\|_2^2)\mathrm{d}\theta$$
$$\leq c_* + c_\sharp + \frac{1}{2}\left(1 + \frac{\lambda_*}{\lambda_\sharp}\right)p\exp(2c_\sharp)$$
$$+ \log\int\exp(c_* - \lambda_*\|\theta\|_2^2)\mathrm{d}\theta - \log\int\exp(-c_\sharp - \lambda_\sharp\|\theta\|_2^2)\mathrm{d}\theta$$
$$\leq 2c_* + 2c_\sharp + \frac{1}{2}\left(1 + \frac{\lambda_*}{\lambda_\sharp}\right)p\exp(2c_\sharp) + \frac{p}{2}\log\left(\frac{\lambda_\sharp}{\lambda_*}\right),$$

where for the first inequality we used a similar inequality as in (12) and for the second inequality we used the Gaussian integral. Hence, we get

$$\mathrm{KL}(q\|q_*) \leq 4c_* + 2c_\sharp + \frac{3}{2}\left(1 + \frac{\lambda_*}{\lambda_\sharp}\right) p \exp(2c_\sharp) + \frac{p}{2}\log\left(\frac{\lambda_\sharp}{\lambda_*}\right)$$
$$+ \left(1 + 4\left(1 + \frac{\lambda_*}{\lambda_\sharp}\right)\exp(4c_\sharp)\right)\mathrm{KL}(q\|q_\sharp) + c_\sharp\sqrt{2\mathrm{KL}(q\|q_\sharp)}.$$

$\square$

**Lemma E.** *Suppose Assumption **(A1')** and **(A2')** hold. If $\mathrm{KL}(q^{(t)}\|q_*^{(t)}) \leq \frac{1}{t^2}$ for $t \geq 2$, then*

$$t\left|\int g^{(t)}(\theta)(q^{(t)}(\theta) - q_*^{(t)}(\theta))\mathrm{d}\theta\right|, \quad \lambda_2 t\left|e(q^{(t)}) - e(q_*^{(t)})\right| = O\left(1 + \lambda_2 + p\lambda_2\exp(8/\lambda_2)\right).$$

*Proof.* Recall the definition of $g^{(t)}, \overline{g}^{(t)}$ and $q_*^{(t)}$ (see notations in subsection B. 1). We set $\gamma_{t+1} = \frac{\sum_{s=1}^{t} s}{\lambda_2 \sum_{s=1}^{t+1} s} = \frac{t}{\lambda_2(t+2)}$. Note that for $t \geq 1$,

$$-2 + \lambda_1 \|\theta\|_2^2 \leq g^{(t)}(\theta) \leq 2 + \lambda_1 \|\theta\|_2^2, \tag{13}$$

$$\gamma_{t+1}(-2 + \lambda_1 \|\theta\|_2^2) \leq \overline{g}^{(t)}(\theta) \leq \gamma_{t+1}(2 + \lambda_1 \|\theta\|_2^2), \tag{14}$$

$$\frac{1}{3\lambda_2} \leq \gamma_{t+1} \leq \frac{1}{\lambda_2}. \tag{15}$$

Therefore, we have for $t \geq 2$ from Proposition A with $\delta = 1/t < 1$,

$$t \left| \int g^{(t)}(\theta)(q^{(t)}(\theta) - q_*^{(t)}(\theta)) \mathrm{d}\theta \right|$$

$$\leq 2t \|q^{(t)} - q_*^{(t)}\|_{L_1(\mathrm{d}\theta)} + \lambda_1 t \left| \int \|\theta\|_2^2 (q^{(t)}(\theta) - q_*^{(t)}(\theta)) \mathrm{d}\theta \right|$$

$$\leq 2t \sqrt{2\mathrm{KL}(q^{(t)}\|q_*^{(t)})}$$
$$\quad + \lambda_2(t+2) \left( (2 + \delta + 1/\delta) \exp(8/\lambda_2)\mathrm{KL}(q^{(t)}\|q_*^{(t)}) + \frac{\delta(1+\delta)p \exp(4/\lambda_2)}{2} \right)$$

$$\leq 2\sqrt{2} + 3\lambda_2 \left( 4\exp(8/\lambda_2) + p \exp(4/\lambda_2) \right)$$
$$= O\left( 1 + p\lambda_2 \exp(8/\lambda_2) \right).$$

Moreover, we have for $t \geq 2$,

$$\lambda_2 t \left| e(q^{(t)}) - e(q_*^{(t)}) \right|$$

$$\leq \lambda_2 t \left( (1 + (2 + \delta + 1/\delta)\exp(8/\lambda_2)) \mathrm{KL}(q^{(t)}\|q_*^{(t)}) + \frac{2}{\lambda_2}\sqrt{2\mathrm{KL}(q^{(t)}\|q_*^{(t)})} + \frac{\delta(1+\delta)p \exp(4/\lambda_2)}{2} \right)$$

$$\leq \lambda_2 t \left( (1 + (3 + t)\exp(8/\lambda_2)) \frac{1}{(t-1)^2} + \frac{2\sqrt{2}}{\lambda_2(t-1)} + \frac{p \exp(4/\lambda_2)}{t} \right)$$

$$= O\left( 1 + \lambda_2 + p\lambda_2 \exp(8/\lambda_2) \right).$$

This finishes the proof. $\qquad\square$

## B. 3 Outer Loop Complexity

Based on the auxiliary results and the convex optimization theory developed in Nesterov (2009); Xiao (2009), we now prove Theorem D which is an extension of Theorem 1.

*Proof of Theorem D.* For $t \geq 1$ we define,

$$V_t(q) = -\mathbb{E}_q \left[ \sum_{s=1}^{t} s g^{(s)} \right] - \lambda_2 e(q) \sum_{s=1}^{t+1} s.$$

From the definition, the density $q_*^{(t+1)} \in \mathcal{P}_2$ calculated in Algorithm 4 maximizes $V_t(q)$. We denote $V_t^* = V(q_*^{(t+1)})$. Then, for $t \geq 2$, we get

$$V_t^* = -\mathbb{E}_{q_*^{(t+1)}} \left[ \sum_{s=1}^{t-1} s g^{(s)} \right] - \lambda_2 e(q_*^{(t+1)}) \sum_{s=1}^{t} s - \mathbb{E}_{q_*^{(t+1)}} \left[ t g^{(t)} \right] - \lambda_2(t+1)e(q_*^{(t+1)})$$

$$\leq V_{t-1}^* - \frac{\lambda_2 \sum_{s=1}^{t} s}{2} \|q_*^{(t+1)} - q_*^{(t)}\|_{L_1(\mathrm{d}\theta)}^2 - \mathbb{E}_{q_*^{(t)}} \left[ t g^{(t)} \right] - \lambda_2(t+1)e(q_*^{(t+1)})$$
$$\quad + t \int (q_*^{(t)} - q_*^{(t+1)})(\theta) g^{(t)}(\theta) \mathrm{d}\theta$$

$$\leq V_{t-1}^* - \frac{\lambda_2 \sum_{s=1}^{t} s}{2} \|q_*^{(t+1)} - q_*^{(t)}\|_{L_1(\mathrm{d}\theta)}^2 - \mathbb{E}_{q^{(t)}} \left[ t g^{(t)} \right] - \lambda_2(t+1)e(q_*^{(t+1)})$$

$$+ t\left|\int g^{(t)}(\theta)(q^{(t)}(\theta) - q_*^{(t)}(\theta))\mathrm{d}\theta\right| + t\int(q_*^{(t)} - q_*^{(t+1)})(\theta)g^{(t)}(\theta)\mathrm{d}\theta$$

$$\leq V_{t-1}^* - \frac{\lambda_2\sum_{s=1}^t s}{2}\|q_*^{(t+1)} - q_*^{(t)}\|_{L_1(\mathrm{d}\theta)}^2 - \mathbb{E}_{q^{(t)}}\left[tg^{(t)}\right] - \lambda_2(t+1)e(q_*^{(t+1)})$$

$$+ t\int(q_*^{(t)} - q_*^{(t+1)})(\theta)g^{(t)}(\theta)\mathrm{d}\theta + O(1 + \lambda_2 + p\lambda_2\exp(8/\lambda_2)), \tag{16}$$

where for the first inequality we used the optimality of $q_*^{(t)}$ and the strong convexity (10) at $q_*^{(t)}$, and for the final inequality we used Lemma E.

We set $R_t = \left(\frac{3}{2}p + 15\right)\frac{\lambda_2}{\lambda_1}\log(1+t)$ and also $\gamma_{t+1} = \frac{\sum_{s=1}^t s}{\lambda_2\sum_{s=1}^{t+1} s} = \frac{t}{\lambda_2(t+2)}$, as done in the proof of Lemma E.

From Assumptions **(A1')**, **(A2')** and $q_*^{(t)} = \exp\left(-\frac{\sum_{s=1}^{t-1} sg^{(s)}}{\lambda_2\sum_{s=1}^t s}\right) / \int \exp\left(-\frac{\sum_{s=1}^{t-1} sg^{(s)}(\theta)}{\lambda_2\sum_{s=1}^t s}\right)\mathrm{d}\theta$ ($t \geq 2$), we have for $t \geq 2$,

$$q_*^{(t)}(\theta) \leq \exp(\gamma_t(2 - \lambda_1\|\theta\|_2^2))/ \int \exp(\gamma_t(-2 - \lambda_1\|\theta\|_2^2))\mathrm{d}\theta$$

$$\leq \exp(4\gamma_t)\exp(-\gamma_t\lambda_1\|\theta\|_2^2)/ \int \exp(-\gamma_t\lambda_1\|\theta\|_2^2)\mathrm{d}\theta$$

$$\leq \exp(4/\lambda_2)\exp(-\gamma_t\lambda_1\|\theta\|_2^2)/ \int \exp(-\gamma_t\lambda_1\|\theta\|_2^2)\mathrm{d}\theta. \tag{17}$$

Using (17) and applying Lemma D with $\sigma^2 = \frac{1}{2\gamma_t\lambda_1}, \frac{1}{2\gamma_{t+1}\lambda_1}$ and $R = R_t$, we have for $t \geq 2$,

$$\left|\int(q_*^{(t)} - q_*^{(t+1)})(\theta)g^{(t)}(\theta)\mathrm{d}\theta\right|$$

$$\leq 2\|q_*^{(t)} - q_*^{(t+1)}\|_{L_1(\mathrm{d}\theta)} + \lambda_1\int\|\theta\|_2^2|(q_*^{(t)} - q_*^{(t+1)})(\theta)|\mathrm{d}\theta$$

$$\leq (2 + 2\lambda_1 R_t)\|q_*^{(t)} - q_*^{(t+1)}\|_{L_1(\mathrm{d}\theta)} + \lambda_1\int_{\|\theta\|_2^2 > 2R_t}\|\theta\|_2^2(q_*^{(t)} + q_*^{(t+1)})(\theta)\mathrm{d}\theta$$

$$\leq (2 + 2\lambda_1 R_t)\|q_*^{(t)} - q_*^{(t+1)}\|_{L_1(\mathrm{d}\theta)} + \lambda_1\exp(4/\lambda_2)\int_{\|\theta\|_2^2 > 2R_t}\|\theta\|_2^2\frac{\exp(-\gamma_t\lambda_1\|\theta\|_2^2)}{\int\exp(-\gamma_t\lambda_1\|\theta\|_2^2)\mathrm{d}\theta}\mathrm{d}\theta$$

$$+ \lambda_1\exp(4/\lambda_2)\int_{\|\theta\|_2^2 > 2R_t}\|\theta\|_2^2\frac{\exp(-\gamma_{t+1}\lambda_1\|\theta\|_2^2)}{\int\exp(-\gamma_{t+1}\lambda_1\|\theta\|_2^2)\mathrm{d}\theta}\mathrm{d}\theta$$

$$\leq (2 + 2\lambda_1 R_t)\|q_*^{(t)} - q_*^{(t+1)}\|_{L_1(\mathrm{d}\theta)} + 2\lambda_1\exp(4/\lambda_2)\left(R_t + \frac{5}{\lambda_1\gamma_t}\right)\exp\left(-\frac{\lambda_1 R_t\gamma_t}{5}\right)$$

$$+ 2\lambda_1\exp(4/\lambda_2)\left(R_t + \frac{5}{\lambda_1\gamma_t}\right)\exp\left(-\frac{\lambda_1 R_t\gamma_{t+1}}{5}\right)$$

$$\leq (2 + 2\lambda_1 R_t)\|q_*^{(t)} - q_*^{(t+1)}\|_{L_1(\mathrm{d}\theta)} + 4\lambda_1\exp(4/\lambda_2)\left(R_t + 15\frac{\lambda_2}{\lambda_1}\right)\exp\left(-\frac{R_t\lambda_1}{15\lambda_2}\right)$$

$$\leq \left(2 + 2\left(\frac{3}{2}p + 15\right)\lambda_2\log(1+t)\right)\|q_*^{(t)} - q_*^{(t+1)}\|_{L_1(\mathrm{d}\theta)} + 8\exp(4/\lambda_2)\left(\frac{3}{2}p + 15\right)\frac{\lambda_2\log(1+t)}{(1+t)^{1+\frac{p}{10}}},$$

where for the fifth inequality we used (15) and for the sixth inequality we used $15\lambda_2/\lambda_1 \leq R_t$.

Applying Young's inequality $ab \leq \frac{a^2}{2\delta} + \frac{\delta b^2}{2}$ with $a = \left(2 + 2\left(\frac{3}{2}p + 15\right)\lambda_2\log(1+t)\right)$, $b = \|q_*^{(t)} - q_*^{(t+1)}\|_{L_1(\mathrm{d}\theta)}$, and $\delta = \frac{\lambda_2}{2}(t+1)$, we get

$$\left|\int(q_*^{(t)} - q_*^{(t+1)})(\theta)g^{(t)}(\theta)\mathrm{d}\theta\right| \leq \frac{\left(2 + 2\left(\frac{3}{2}p + 15\right)\lambda_2\log(1+t)\right)^2}{\lambda_2(t+1)} + \frac{\lambda_2(t+1)\|q_*^{(t)} - q_*^{(t+1)}\|_{L_1(\mathrm{d}\theta)}^2}{4}$$

$$+ 8\exp(4/\lambda_2)\left(\frac{3}{2}p + 15\right)\frac{\lambda_2\log(1+t)}{(1+t)^{1+\frac{p}{10}}}. \tag{18}$$

Combining (16) and (18), we have for $t \geq 2$,

$$
\begin{aligned}
V_t^* &\leq V_{t-1}^* - \mathbb{E}_{q^{(t)}}\left[tg^{(t)}\right] - \lambda_2(t+1)e(q_*^{(t+1)}) + O(1 + \lambda_2 + p\lambda_2 \exp(8/\lambda_2)) \\
&\quad + \frac{1}{\lambda_2}\left(2 + 2\left(\frac{3}{2}p + 15\right)\lambda_2 \log(1+t)\right)^2 + 8\exp(4/\lambda_2)\left(\frac{3}{2}p + 15\right)\frac{\lambda_2 \log(1+t)}{(1+t)^{\frac{p}{10}}} \\
&= V_{t-1}^* - \mathbb{E}_{q^{(t)}}\left[tg^{(t)}\right] - \lambda_2(t+1)e(q_*^{(t+1)}) + O(1 + \lambda_2 + p\lambda_2 \exp(8/\lambda_2)) \\
&\quad + O\left(\frac{1}{\lambda_2} + p^2\lambda_2 \log^2(1+t) + p\lambda_2 \exp(4/\lambda_2)\right) \\
&= V_{t-1}^* - \mathbb{E}_{q^{(t)}}\left[tg^{(t)}\right] - \lambda_2(t+1)e(q_*^{(t+1)}) + O\left(p\lambda_2 \exp(8/\lambda_2) + p^2\lambda_2 \log^2(1+t)\right) \\
&= V_{t-1}^* - \mathbb{E}_{q^{(t)}}\left[tg^{(t)}\right] - \lambda_2(t+1)e(q_*^{(t+1)}) + O\left((1 + \exp(8/\lambda_2))p^2\lambda_2 \log^2(1+t)\right) \\
&= V_{t-1}^* - \mathbb{E}_{q^{(t)}}\left[tg^{(t)}\right] - \lambda_2(t+1)e(q_*^{(t+1)}) + \alpha_t, \quad (19)
\end{aligned}
$$

where we set $\alpha_t = O\left((1 + \exp(8/\lambda_2))p^2\lambda_2 \log^2(1+t)\right)$.

From Proposition B, (14), and (15),

$$
-\mathbb{E}_{q_*^{(t)}}[\log(q_*^{(t)})] \leq \frac{4}{\lambda_2} + \frac{p}{2}\left(\exp\left(\frac{4}{\lambda_2}\right) + \log\left(\frac{3\pi\lambda_2}{\lambda_1}\right)\right),
$$

meaning $e(q_*^{(t)}) \geq 0$. Hence,

$$
V_1^* = -\mathbb{E}_{q_*^{(2)}}[g^{(1)}] - 3\lambda_2 e(q_*^{(2)}) \leq 2 - 3\lambda_2 e(q_*^{(2)}) \leq 2 - 2\lambda_2 e(q_*^{(2)}).
$$

Summing the inequality (19) over $t \in \{2, \ldots, T+1\}$,

$$
\begin{aligned}
V_{T+1}^* &\leq 2 - 2\lambda_2 e(q_*^{(2)}) + \sum_{t=2}^{T+1}\left\{-\mathbb{E}_{q^{(t)}}\left[tg^{(t)}\right] - \lambda_2(t+1)e(q_*^{(t+1)}) + \alpha_t\right\} \\
&= 2 - \sum_{t=2}^{T+1}t\left\{\mathbb{E}_{q^{(t)}}\left[g^{(t)}\right] + \lambda_2 e(q_*^{(t)})\right\} + \sum_{t=2}^{T+1}\alpha_t - \lambda_2(T+2)e(q_*^{(T+2)}) \\
&\leq 2 - \sum_{t=2}^{T+1}t\left\{\mathbb{E}_{q^{(t)}}\left[g^{(t)}\right] + \lambda_2 e(q^{(t)})\right\} + \sum_{t=2}^{T+1}\alpha_t, \quad (20)
\end{aligned}
$$

where we used $\lambda_2 t \left|e(q^{(t)}) - e(q_*^{(t)})\right| = \alpha_t$ (Lemma E), $2\alpha_t = O(\alpha_t)$, and $e(q_*^{(T+2)}) \geq 0$.

On the other hand, for $\forall q_* \in \mathcal{P}_2$,

$$
V_{T+1}^* = \max_{q \in \mathcal{P}_2}\left\{-\mathbb{E}_q\left[\sum_{t=1}^{T+1}tg^{(t)}\right] - \lambda_2 e(q)\sum_{t=1}^{T+2}t\right\} \geq -\mathbb{E}_{q_*}\left[\sum_{t=1}^{T+1}tg^{(t)}\right] - \lambda_2 e(q_*)\sum_{t=1}^{T+2}t. \quad (21)
$$

Using **(A1')**, **(A2')**, and **(A3')**, we have for any density function $q$,

$$
\left|(\partial_z\ell(h_{q^{(t)}}(x_t), y_t) - \partial_z\ell(h_x^{(t)}, y_t))\mathbb{E}_q[h(\cdot, x_t)]\right| \leq \epsilon. \quad (22)
$$

Hence, from (20), (21), (22), and the convexity of the loss,

$$
\frac{2}{T(T+3)} \sum_{t=2}^{T+1} t \Big\{ \ell(h_{q^{(t)}}(x_t), y_t) + \lambda_1 \mathbb{E}_{q^{(t)}}[\|\theta\|_2^2] + \lambda_2 \mathbb{E}_{q^{(t)}}[\log(q^{(t)})]
$$

$$
- \ell(h_{q_*}(x_t), y_t) - \lambda_1 \mathbb{E}_{q_*}[\|\theta\|_2^2] - \lambda_2 \mathbb{E}_{q_*}[\log(q_*)] \Big\}
$$

$$
\leq \frac{2}{T(T+3)} \sum_{t=2}^{T+1} t \Big\{ \partial_z \ell(h_{q^{(t)}}(x_t), y_t) \left( \mathbb{E}_{q^{(t)}}[h(\cdot, x_t)] - \mathbb{E}_{q_*}[h(\cdot, x_t)] \right)
$$

$$
+ \lambda_1 \left( \mathbb{E}_{q^{(t)}}[\|\theta\|_2^2] - \mathbb{E}_{q_*}[\|\theta\|_2^2] \right) + \lambda_2 \left( \mathbb{E}_{q^{(t)}}[\log(q^{(t)})] - \mathbb{E}_{q_*}[\log(q_*)] \right) \Big\}
$$

$$
\leq \frac{2}{T(T+3)} \sum_{t=2}^{T+1} t \left\{ 2\epsilon + \mathbb{E}_{q^{(t)}}[g^{(t)}] - \mathbb{E}_{q_*}[g^{(t)}] + \lambda_2 \left( e(q^{(t)}) - e(q_*) \right) \right\}
$$

$$
\leq 2\epsilon + \frac{2}{T(T+3)} \left( 2 - V_{T+1}^* + \sum_{t=2}^{T+1} \alpha_t - \sum_{t=2}^{T+1} t \left( \mathbb{E}_{q_*}[g^{(t)}] + \lambda_2 e(q_*) \right) \right)
$$

$$
\leq 2\epsilon + \frac{2}{T(T+3)} \left( 2 + \mathbb{E}_{q_*}\left[ g^{(1)} \right] + \lambda_2 (T+3) e(q_*) + \sum_{t=2}^{T+1} \alpha_t \right)
$$

$$
\leq 2\epsilon + \frac{2}{T(T+3)} \left( 4 + \lambda_1 \mathbb{E}_{q_*}\left[ \|\theta\|_2^2 \right] \right) + \frac{2\lambda_2 e(q_*)}{T} + \frac{2}{T} O \left( (1 + \exp(8/\lambda_2)) p^2 \lambda_2 \log^2(T+2) \right).
$$

Taking the expectation with respect to the history of examples, we have

$$
\frac{2}{T(T+3)} \sum_{t=2}^{T+1} t \left( \mathbb{E}[\mathcal{L}(q^{(t)})] - \mathcal{L}(q_*) \right)
$$

$$
= 2\epsilon + O \left( \frac{1}{T^2} \left( 1 + \lambda_1 \mathbb{E}_{q_*}\left[ \|\theta\|_2^2 \right] \right) + \frac{\lambda_2}{T} \left( e(q_*) + (1 + \exp(8/\lambda_2)) p^2 \log^2(T+2) \right) \right).
$$

$\square$

## B.4 Inner Loop Complexity

We next prove Corollary 1 which gives an estimate of inner loop iteration complexity. This result is derived by utilizing the convergence rate of the Langevin algorithm under LSI developed in Vempala and Wibisono (2019). We here consider the ideal Algorithm 2 (i.e., warm-start and exact mean field limit ($\epsilon = 0$)).

*Proof of Corollary 1.* We verify the assumptions required in Theorem 2. We recall that $q_*^{(t+1)}$ takes the form of Boltzmann distribution: for $t \geq 1$,

$$
q_*^{(t+1)} \propto \exp\left( - \frac{\sum_{s=1}^t s g^{(s)}}{\lambda_2 \sum_{s=1}^{t+1} s} \right)
$$

$$
= \exp\left( - \frac{1}{\lambda_2 \sum_{s=1}^{t+1} s} \sum_{s=1}^t s \partial_z \ell(h_x^{(t)}, y_t) h(\cdot, x_t) - \frac{\lambda_1 t}{\lambda_2 (t+2)} \|\theta\|_2^2 \right).
$$

Note that $\frac{\lambda_1}{\lambda_2} \geq \frac{\lambda_1 t}{\lambda_2(t+2)} \geq \frac{\lambda_1}{3\lambda_2}$ ($t \geq 1$) and $\left| \frac{1}{\lambda_2 \sum_{s=1}^{t+1} s} \sum_{s=1}^t s \partial_z \ell(h_x^{(t)}, y_t) h(\cdot, x_t) \right| \leq \frac{2t}{\lambda_2(t+2)} \leq \frac{2}{\lambda_2}$. Therefore, from Example 2 and Lemma B, we know that $q_*^{(t+1)}$ satisfies the log-Sobolev inequality with a constant $\frac{2\lambda_1}{3\lambda_2 \exp(8/\lambda_2)}$; in addition, the gradient of $\log(q_*^{(t+1)})$ is $\frac{2}{\lambda_2}(1 + \lambda_1)$-Lipschitz continuous. Therefore, from Theorem 2 we deduce that Langevin algorithm with learning rate $\eta_t \leq \frac{\lambda_1 \lambda_2 \delta_{t+1}}{96p(1+\lambda_1)^2 \exp(8/\lambda_2)}$ yields $q^{t+1}$ satisfying $\mathrm{KL}(q^{(t+1)} \| q_*^{(t+1)}) \leq \delta_{t+1}$ within $\frac{3\lambda_2 \exp(8/\lambda_2)}{2\lambda_1 \eta_t} \log \frac{2\mathrm{KL}(q^{(t)} \| q_*^{(t+1)})}{\delta_{t+1}}$-iterations.

We next bound $\mathrm{KL}(q^{(t)}\|q_*^{(t+1)})$. Apply Proposition C with $q = q^{(t)}$, $q_* = q_*^{(t+1)}$, and $q_\sharp = q_*^{(t)}$. Note that in this setting, constants $c_*, c_\sharp, \lambda_*$, and $\lambda_\sharp$ satisfy

$$c_* \leq \frac{2}{\lambda_2}, \quad \frac{\lambda_1}{3\lambda_2} \leq \lambda_* \leq \frac{\lambda_1}{\lambda_2},$$

$$c_\sharp \leq \frac{2}{\lambda_2}, \quad \frac{\lambda_1}{3\lambda_2} \leq \lambda_\sharp \leq \frac{\lambda_1}{\lambda_2}.$$

Then, we get

$$\mathrm{KL}(q^{(t)}\|q_*^{(t+1)}) \leq \frac{12}{\lambda_2} + 6p \exp\left(\frac{4}{\lambda_2}\right) + \frac{p}{2}\log 3 + \left(1 + 16\exp\left(\frac{8}{\lambda_2}\right)\right)\mathrm{KL}(q^{(t)}\|q_*^{(t)})$$
$$+ \frac{2}{\lambda_2}\sqrt{2\mathrm{KL}(q^{(t)}\|q_*^{(t)})}.$$

Hence, we can conclude $\mathrm{KL}(q^{(t)}\|q_*^{(t+1)})$ are uniformly bounded with respect to $t \in \{1, \ldots, T\}$ as long as $\mathrm{KL}(q^{(t)}\|q_*^{(t)}) \leq \delta_t$ and $q^{(1)}$ is a Gaussian distribution. $\qquad\square$

**Case of resampling.** We note that for resampling scheme, the similar inner loop complexity of $O\left(\frac{\lambda_2 \exp(8/\lambda_2)}{\lambda_1 \eta_t}\log\frac{2\mathrm{KL}(q^{(1)}\|q_*^{(t+1)})}{\delta_{t+1}}\right)$ can be immediately obtained by replacing the initial distribution of Langevin algorithm with $q^{(1)}(\theta)\mathrm{d}\theta$. Moreover, the uniform boundedness of $\mathrm{KL}(q^{(1)}\|q_*^{(t+1)})$ with respect to $t$ is also guaranteed by applying Proposition C with $q = q_\sharp = q^{(1)}$ and $q_* = q_*^{(t+1)}$ as long as $q^{(1)}(\theta)\mathrm{d}\theta$ is a Gaussian distribution.

## ADDITIONAL RESULTS AND DISCUSSIONS

## C  Discretization Error of Finite Particles

### C. 1  Case of Resampling

As discussed in subsection B. 1, to establish the finite-particle convergence guarantees of Algorithm 3 with resampling up to $O(\epsilon)$-error, we need to show that $h_x^{(t)} = h_{\tilde{\Theta}^{(t)}}(x_t)$ satisfies the condition $|h_x^{(t)} - h_{q^{(t)}}(x_t)| \leq \epsilon$ in **(A3')**. Hence, we are interested in characterizing the discretization error that stems from using finitely many particles.

For the resampling scheme, we can easily derive that the required number of particles is $O(\epsilon^{-2}\log(T/\delta))$ with high probability $1 - \delta$, because i.i.d. particles are obtained by the Langevin algorithm and Hoeffding's inequality is applicable.

**Lemma F** (Hoeffding's inequality). *Let $Z, Z_1, \ldots, Z_m$ be i.i.d. random variables taking values in $[-a, a]$ for $a > 0$. Then, for any $\rho > 0$, we get*

$$\mathbb{P}\left[\left|\frac{1}{M}\sum_{r=1}^{M}Z_r - \mathbb{E}[Z]\right| > \rho\right] \leq 2\exp\left(-\frac{\rho^2 M}{2a^2}\right).$$

### C. 2  Case of Warm-start

We next consider the warm-start scheme. Note that the convergence of PDA with warm-start is guaranteed by coupling it with its mean-field limit $M \to \infty$ and applying Theorem 1 without tolerance (i.e., $\epsilon = 0$). To analyze the particle complexity, we make an additional assumption regarding the regularity of the loss function and the model.

**Assumption D.**

  **(A5)** $h(\cdot, x)$ is $1$-Lipschitz continuous[6] for $\forall x \in \mathcal{X}$.

---
[6]WLOG the Lipschitz constant is set to 1, since the same analysis works for any fixed constant.

**Remark.** The above regularity assumption is common in the literature and cover many important problem settings in the optimization of two-layer neural network in the mean field regime. Indeed, **(A5)** is satisfied for two-layer network in Example 1 when the output or input layer is fixed and when the activation function is Lipschitz continuous.

The following proposition shows the convergence of Algorithm 1 to Algorithm 2 as $M \to \infty$.

**Proposition D** (Finite Particle Approximation). *For training examples $\{x_t\}_{t=1}^T$ and any example $\tilde{x}$, define*

$$\rho_{T,M} = \max_{\substack{s \in \{1,\dots,T\} \\ t \in \{1,\dots,T+1\}}} \left| h_{q^{(t)}}(x_s) - h_{\tilde{\Theta}^{(t)}}(x_s) \right| \vee \left| h_{q^{(t)}}(\tilde{x}) - h_{\tilde{\Theta}^{(t)}}(\tilde{x}) \right|.$$

*Under **(A1')**, **(A2)**, **(A4)**, and **(A5)**, if we run PDA (Algorithm 1) on $\tilde{\Theta}$ and the corresponding mean field limit DA (Algorithm 2) on $q$, then with high probability $\lim_{M \to \infty} \rho_{T,M} = 0$. Moreover, if we set $\eta_t \leq \frac{\lambda_2}{2\lambda_1}$, $\lambda_1 \geq \frac{3}{2}$, and $T_t \geq \frac{3\lambda_2 \log(4)}{(2\lambda_1 - 1)\eta_t}$, then with probability at least $1 - \delta$,*

$$\rho_{T,M} \leq \left( 1 + \frac{4}{2\lambda_1 - 1} \right) \sqrt{\frac{2}{M} \log \left( \frac{2(T+1)^2}{\delta} \right)}.$$

**Remark.** Proposition D together with Corollary 2 imply that under appropriate regularization, a prediction on any point with an $\epsilon$-gap from an $\epsilon$-accurate solution of the regularized objective (4) can be achieved with high probability by running PDA with warm-start (Algorithm 1) in $\mathrm{poly}(\epsilon^{-1})$ steps using $\mathrm{poly}(\epsilon^{-1})$ particles, where we omit dependence on hyperparameters and logarithmic factors. Note that specific choices of hyper-parameters in Proposition D are consistent with those in Corollary 2. We also remark that under weak regularization (vanishing $\lambda_1$), our current derivation suggests that the required particle size could be exponential in the time horizon, due to the particle correlation in the warm-start scheme. Finally, we remark that for the empirical risk minimization, the term $\log(2(T+1)^2/\delta)$ could be changed to $\log(2n(T+1)/\delta)$ in the obvious way.

*Proof of Proposition D.* We analyze an error of finite particle approximation for a fixed history of data $\{x_t\}_{t=1}^T$. To Algorithm 2 with the corresponding particle dynamics (Algorithm 1), we construct an *semi particle dual averaging* update, which is an intermediate of these two algorithms. In particular, the semi particle dual averaging method is defined by replacing $h_{\tilde{\Theta}^{(t)}}$ in Algorithm 1 with $h_{q^{(t)}}$ for $q^{(t)}$ in Algorithm 2. Let $\tilde{\Theta}'^{(t)} = \{\tilde{\theta}_r'^{(t)}\}_{r=1}^M$ be parameters obtained in outer loop of the semi particle dual averaging. We first estimate the gap between Algorithm 2 and the semi particle dual averaging.

Note that there is no interaction among $\tilde{\Theta}'^{(t)}$; in other words these are i.i.d. particles sampled from $q^{(t)}$, and we can thus apply Hoeffding's inequality (Lemma F) to $h_{\tilde{\Theta}'^{(t)}}(\tilde{x})$ and $h_{\tilde{\Theta}'^{(t)}}(x_s)$ ($s \in \{1,\dots,T\}, t \in \{1,\dots,T+1\}$). Hence, for $\forall \delta > 0$, $\forall s \in \{1,\dots,T\}$, and $\forall t \in \{1,\dots,T+1\}$, with the probability at least $1 - \delta$

$$\left| h_{\tilde{\Theta}'^{(t)}}(x_s) - h_{q^{(t)}}(x_s) \right| = \left| \frac{1}{M} \sum_{r=1}^M h_{\tilde{\theta}_r'^{(t)}}(x_s) - h_{q^{(t)}}(x_s) \right| \leq \sqrt{\frac{2}{M} \log \left( \frac{2(T+1)^2}{\delta} \right)}, \quad (23)$$

$$\left| h_{\tilde{\Theta}'^{(t)}}(\tilde{x}) - h_{q^{(t)}}(\tilde{x}) \right| = \left| \frac{1}{M} \sum_{r=1}^M h_{\tilde{\theta}_r'^{(t)}}(\tilde{x}) - h_{q^{(t)}}(\tilde{x}) \right| \leq \sqrt{\frac{2}{M} \log \left( \frac{2(T+1)^2}{\delta} \right)}. \quad (24)$$

We next bound the gap between the semi particle dual averaging and Algorithm 1 sharing a history of Gaussian noises and initial particles. That is, $\tilde{\theta}_r^{(1)} = \tilde{\theta}_r'^{(1)}$. Let $\Theta^{(k)} = \{\theta_r^{(k)}\}_{r=1}$ and $\Theta'^{(k)} = \{\theta_r'^{(k)}\}_{r=1}$ denote inner iterations of these methods.

(*i*) Here we show the first statement of the proposition. We set $\rho_1 = 0$ and $\bar{\rho}_1 = 0$. We define $\rho_t$ and $\bar{\rho}_t$ recursively as follows.

$$\rho_{t+1} \overset{\text{def}}{=} \left( 1 + \frac{2(1+\lambda_1)t\eta_t}{\lambda_2(t+2)} \right)^{T_t} \bar{\rho}_t$$

$$+ \frac{t\eta_t}{\lambda_2(t+2)} \left( \bar{\rho}_t + \sqrt{\frac{2}{M} \log \left( \frac{2(T+1)^2}{\delta} \right)} \right) \sum_{s=0}^{T_t - 1} \left( 1 + \frac{2(1+\lambda_1)t\eta_t}{\lambda_2(t+2)} \right)^s, \quad (25)$$

and $\bar{\rho}_{t+1} = \max_{s \in \{1,\ldots,t+1\}} \rho_s$. We show that for any event where (23) and (24) hold, $\|\tilde{\theta}_r^{(t)} - \tilde{\theta}_r'^{(t)}\|_2 \leq \rho_t$ ($\forall t \in \{1,\ldots,T+1\}, \forall r \in \{1,\ldots,M\}$) by induction. Suppose $\|\tilde{\theta}_r^{(s)} - \tilde{\theta}_r'^{(s)}\|_2 \leq \rho_s$ ($\forall s \in \{1,\ldots,t\}, \forall r \in \{1,\ldots,M\}$) holds. Then, for any $x$ and $s \in \{1,\ldots,t\}$

$$|h_{\tilde{\Theta}^{(s)}}(x) - h_{\tilde{\Theta}'^{(s)}}(x)| \leq \frac{1}{M} \sum_{r=1}^M \left| h(\tilde{\theta}_r^{(s)}, x) - h(\tilde{\theta}_r'^{(s)}, x) \right|$$

$$\leq \frac{1}{M} \sum_{r=1}^M \left\| \tilde{\theta}_r^{(s)} - \tilde{\theta}_r'^{(s)} \right\|_2 \leq \rho_s. \tag{26}$$

Consider the inner loop at $t$-the outer step. Then, for an event where (23) holds,

$\|\theta_r^{(k+1)} - \theta_r'^{(k+1)}\|_2$

$\leq \left\| \theta_r^{(k)} - \frac{2\eta_t}{\lambda_2(t+2)(t+1)} \sum_{s=1}^t s \left( \partial_z \ell(h_{\tilde{\Theta}^{(s)}}(x_s), y_s) \partial_\theta h(\theta_r^{(k)}, x_s) + 2\lambda_1 \theta_r^{(k)} \right) \right.$

$\left. - \theta_r'^{(k)} + \frac{2\eta_t}{\lambda_2(t+2)(t+1)} \sum_{s=1}^t s \left( \partial_z \ell(h_{q^{(s)}}(x_s), y_s) \partial_\theta h(\theta_r'^{(k)}, x_s) + 2\lambda_1 \theta_r'^{(k)} \right) \right\|_2$

$\leq \left( 1 + \frac{2\lambda_1 t\eta_t}{\lambda_2(t+2)} \right) \|\theta_r^{(k)} - \theta_r'^{(k)}\|_2$

$+ \frac{2\eta_t}{\lambda_2(t+2)(t+1)} \sum_{s=1}^t s \|\partial_z \ell(h_{\tilde{\Theta}^{(s)}}(x_s), y_s) \partial_\theta h(\theta_r^{(k)}, x_s) - \partial_z \ell(h_{q^{(s)}}(x_s), y_s) \partial_\theta h(\theta_r'^{(k)}, x_s)\|_2$

$\leq \left( 1 + \frac{2\lambda_1 t\eta_t}{\lambda_2(t+2)} \right) \|\theta_r^{(k)} - \theta_r'^{(k)}\|_2$

$+ \frac{2\eta_t}{\lambda_2(t+2)(t+1)} \sum_{s=1}^t s \left\| (\partial_z \ell(h_{\tilde{\Theta}^{(s)}}(x_s), y_s) - \partial_z \ell(h_{q^{(s)}}(x_s), y_s)) \partial_\theta h(\theta_r^{(k)}, x_s) \right\|_2$

$+ \frac{2\eta_t}{\lambda_2(t+2)(t+1)} \sum_{s=1}^t s \left\| \partial_z \ell(h_{q^{(s)}}(x_s), y_s)(\partial_\theta h(\theta_r'^{(k)}, x_s) - \partial_\theta h(\theta_r^{(k)}, x_s)) \right\|_2$

$\leq \left( 1 + \frac{2(1+\lambda_1)t\eta_t}{\lambda_2(t+2)} \right) \|\theta_r^{(k)} - \theta_r'^{(k)}\|_2 + \frac{2\eta_t}{\lambda_2(t+2)(t+1)} \sum_{s=1}^t s \left| h_{\tilde{\Theta}^{(s)}}(x_s) - h_{q^{(s)}}(x_s) \right|$

$\leq \left( 1 + \frac{2(1+\lambda_1)t\eta_t}{\lambda_2(t+2)} \right) \|\theta_r^{(k)} - \theta_r'^{(k)}\|_2 + \frac{2\eta_t}{\lambda_2(t+2)(t+1)} \sum_{s=1}^t s \left( \rho_s + \sqrt{\frac{2}{M} \log \left( \frac{2(T+1)^2}{\delta} \right)} \right)$

$\leq \left( 1 + \frac{2(1+\lambda_1)t\eta_t}{\lambda_2(t+2)} \right) \|\theta_r^{(k)} - \theta_r'^{(k)}\|_2 + \frac{t\eta_t}{\lambda_2(t+2)} \left( \bar{\rho}_t + \sqrt{\frac{2}{M} \log \left( \frac{2(T+1)^2}{\delta} \right)} \right).$

Expanding this inequality,

$\|\tilde{\theta}_r^{(t+1)} - \tilde{\theta}_r'^{(t+1)}\|_2$

$\leq \left( 1 + \frac{2(1+\lambda_1)t\eta_t}{\lambda_2(t+2)} \right)^{T_t} \bar{\rho}_t + \frac{t\eta_t}{\lambda_2(t+2)} \left( \bar{\rho}_t + \sqrt{\frac{2}{M} \log \left( \frac{2(T+1)^2}{\delta} \right)} \right) \sum_{s=0}^{T_t-1} \left( 1 + \frac{2(1+\lambda_1)t\eta_t}{\lambda_2(t+2)} \right)^s$

$= \rho_{t+1}.$

Hence, $\|\tilde{\theta}_r^{(t)} - \tilde{\theta}_r'^{(t)}\|_2 \leq \bar{\rho}_{T+1}$ for $\forall t \in \{1,\ldots,T+1\}$.

Noting that $\bar{\rho}_1 = 0$ and

$$\rho_{t+1} = \left( \left( 1 + \frac{2(1+\lambda_1)t\eta_t}{\lambda_2(t+2)} \right)^{T_t} + \frac{t\eta_t}{\lambda_2(t+2)} \sum_{s=0}^{T_t-1} \left( 1 + \frac{2(1+\lambda_1)t\eta_t}{\lambda_2(t+2)} \right)^s \right) \bar{\rho}_t$$

$$+ \frac{t\eta_t}{\lambda_2(t+2)} \sqrt{\frac{2}{M} \log\left(\frac{2(T+1)^2}{\delta}\right)} \sum_{s=0}^{T_t-1} \left(1 + \frac{2(1+\lambda_1)t\eta_t}{\lambda_2(t+2)}\right)^s,$$

we see $\overline{\rho}_{T+1} \to 0$ as $M \to +\infty$. Then, the proof is finished because for $\forall t \in \{1, \ldots, T+1\}$ and $\forall s \in \{1, \ldots, T\}$ with high probability $1 - \delta$,

$$\left|h_{\tilde{\Theta}^{(t)}}(x_s) - h_{q^{(t)}}(x_s)\right| \le \left|h_{\tilde{\Theta}^{(t)}}(x_s) - h_{\tilde{\Theta}'^{(t)}}(x_s)\right| + \left|h_{\tilde{\Theta}'^{(t)}}(x_s) - h_{q^{(t)}}(x_s)\right|$$

$$\le \overline{\rho}_{T+1} + \sqrt{\frac{2}{M} \log\left(\frac{2(T+1)^2}{\delta}\right)},$$

$$\left|h_{\tilde{\Theta}^{(t)}}(\tilde{x}) - h_{q^{(t)}}(\tilde{x})\right| \le \left|h_{\tilde{\Theta}^{(t)}}(\tilde{x}) - h_{\tilde{\Theta}'^{(t)}}(\tilde{x})\right| + \left|h_{\tilde{\Theta}'^{(t)}}(\tilde{x}) - h_{q^{(t)}}(\tilde{x})\right|$$

$$\le \overline{\rho}_{T+1} + \sqrt{\frac{2}{M} \log\left(\frac{2(T+1)^2}{\delta}\right)}.$$

($ii$) We next show the second statement of the proposition. We change the definition (25) of $\rho_{t+1}$ as follows:

$$\rho_{t+1} \stackrel{\text{def}}{=} \frac{3}{4}\overline{\rho}_t + \frac{1}{2\lambda_1 - 1} \sqrt{\frac{2}{M} \log\left(\frac{2(T+1)^2}{\delta}\right)}.$$

We prove that for any event where (23) and (24) hold, $\|\tilde{\theta}_r^{(t)} - \tilde{\theta}_r'^{(t)}\|_2 \le \rho_t$ ($\forall t \in \{1, \ldots, T+1\}$, $\forall r \in \{1, \ldots, M\}$) by induction. Suppose $\|\tilde{\theta}_r^{(s)} - \tilde{\theta}_r'^{(s)}\|_2 \le \rho_s$ ($\forall s \in \{1, \ldots, t\}$, $\forall r \in \{1, \ldots, M\}$) holds. Consider the inner loop at $t$-step. Note that $\eta_t \le \frac{\lambda_2}{2\lambda_1}$ implies $1 - \frac{2\lambda_1 t\eta_t}{\lambda_2(t+2)} > 0$. Therefore, by the similar argument as above, we get

$$\|\theta_r^{(k+1)} - \theta_r'^{(k+1)}\|_2$$

$$\le \left\| \theta_r^{(k)} - \frac{2\eta_t}{\lambda_2(t+2)(t+1)} \sum_{s=1}^{t} s\left(\partial_z \ell(h_{\tilde{\Theta}^{(s)}}(x_s), y_s)\partial_\theta h(\theta_r^{(k)}, x_s) + 2\lambda_1 \theta_r^{(k)}\right) \right.$$

$$\left. - \theta_r^{(k)} + \frac{2\eta_t}{\lambda_2(t+2)(t+1)} \sum_{s=1}^{t} s\left(\partial_z \ell(h_{q^{(s)}}(x_s), y_s)\partial_\theta h(\theta_r'^{(k)}, x_s) + 2\lambda_1 \theta_r'^{(k)}\right) \right\|_2$$

$$\le \left(1 - \frac{2\lambda_1 t\eta_t}{\lambda_2(t+2)}\right) \|\theta_r^{(k)} - \theta_r'^{(k)}\|_2$$

$$+ \frac{2\eta_t}{\lambda_2(t+2)(t+1)} \sum_{s=1}^{t} s\|\partial_z \ell(h_{\tilde{\Theta}^{(s)}}(x_s), y_s)\partial_\theta h(\theta_r^{(k)}, x_s) - \partial_z \ell(h_{q^{(s)}}(x_s), y_s)\partial_\theta h(\theta_r'^{(k)}, x_s)\|_2$$

$$\le \left(1 + \frac{(1 - 2\lambda_1)t\eta_t}{\lambda_2(t+2)}\right) \|\theta_r^{(k)} - \theta_r'^{(k)}\|_2 + \frac{t\eta_t}{\lambda_2(t+2)} \left(\overline{\rho}_t + \sqrt{\frac{2}{M} \log\left(\frac{2(T+1)^2}{\delta}\right)}\right).$$

Expanding this inequality,

$$\|\tilde{\theta}_r^{(t+1)} - \tilde{\theta}_r'^{(t+1)}\|_2$$

$$\le \left(1 + \frac{(1 - 2\lambda_1)t\eta_t}{\lambda_2(t+2)}\right)^{T_t} \overline{\rho}_t + \frac{t\eta_t}{\lambda_2(t+2)} \left(\overline{\rho}_t + \sqrt{\frac{2}{M} \log\left(\frac{2(T+1)^2}{\delta}\right)}\right) \sum_{s=0}^{T_t-1} \left(1 + \frac{(1 - 2\lambda_1)t\eta_t}{\lambda_2(t+2)}\right)^s$$

$$\le \left(\left(1 + \frac{(1 - 2\lambda_1)t\eta_t}{\lambda_2(t+2)}\right)^{T_t} + \frac{1}{2\lambda_1 - 1}\right) \overline{\rho}_t + \frac{1}{2\lambda_1 - 1} \sqrt{\frac{2}{M} \log\left(\frac{2(T+1)^2}{\delta}\right)}$$

$$\le \left(\left(1 + \frac{(1 - 2\lambda_1)t\eta_t}{\lambda_2(t+2)}\right)^{T_t} + \frac{1}{2}\right) \overline{\rho}_t + \frac{1}{2\lambda_1 - 1} \sqrt{\frac{2}{M} \log\left(\frac{2(T+1)^2}{\delta}\right)},$$

where we used $0 < 1 + \frac{(1-2\lambda_1)t\eta_t}{\lambda_2(t+2)} < 1$ and $\lambda_1 \ge \frac{3}{2}$.

Noting that $(1-x)^{1/x} \leq \exp(-1)$ for $\forall x \in (0,1]$, we see that

$$\left(1 - \frac{(2\lambda_1 - 1)t\eta_t}{\lambda_2(t+2)}\right)^{T_t} \leq \left(1 - \frac{(2\lambda_1 - 1)t\eta_t}{\lambda_2(t+2)}\right)^{\frac{3\lambda_2}{(2\lambda_1 - 1)\eta_t}\log(4)}$$

$$= \left(1 - \frac{(2\lambda_1 - 1)t\eta_t}{\lambda_2(t+2)}\right)^{\frac{\lambda_2(t+2)}{(2\lambda_1 - 1)t\eta_t}\frac{3t}{t+2}\log(4)}$$

$$\leq \exp\left(-\frac{3t}{t+2}\log(4)\right)$$

$$\leq \exp\left(-\log(4)\right)$$

$$= \frac{1}{4},$$

where we used $T_t \geq \frac{3\lambda_2\log(4)}{(2\lambda_1 - 1)\eta_t}$. Hence, we know that for $t$,

$$\|\tilde{\theta}_r^{(t+1)} - \tilde{\theta}_r'^{(t+1)}\|_2 \leq \frac{3}{4}\overline{\rho}_t + \frac{1}{2\lambda_1 - 1}\sqrt{\frac{2}{M}\log\left(\frac{2(T+1)^2}{\delta}\right)}. \tag{27}$$

This means that $\|\tilde{\theta}_r^{(t+1)} - \tilde{\theta}_r'^{(t+1)}\|_2 \leq \rho_{t+1}$ and finishes the induction.

Next, we show

$$\overline{\rho}_t \leq \frac{4}{2\lambda_1 - 1}\sqrt{\frac{2}{M}\log\left(\frac{2(T+1)^2}{\delta}\right)}. \tag{28}$$

This inequality obviously holds for $t = 1$ because $\overline{\rho}_1 = 0$. We suppose it is true for $t \leq T$. Then,

$$\rho_{t+1} = \frac{3}{4}\overline{\rho}_t + \frac{1}{2\lambda_1 - 1}\sqrt{\frac{2}{M}\log\left(\frac{2(T+1)^2}{\delta}\right)}$$

$$\leq \frac{4}{2\lambda_1 - 1}\sqrt{\frac{2}{M}\log\left(\frac{2(T+1)^2}{\delta}\right)}.$$

Hence, the inequality (28) holds for $\forall t \in \{1, \ldots, T+1\}$, yielding

$$\|\tilde{\theta}_r^{(t+1)} - \tilde{\theta}_r'^{(t+1)}\|_2 \leq \frac{4}{2\lambda_1 - 1}\sqrt{\frac{2}{M}\log\left(\frac{2(T+1)^2}{\delta}\right)}.$$

In summary, it follows that for $\forall t \in \{1, \ldots, T+1\}$ and $\forall s \in \{1, \ldots, T\}$ with high probability $1 - \delta$,

$$\left|h_{\tilde{\Theta}^{(t)}}(x_s) - h_{q^{(t)}}(x_s)\right| \leq \left|h_{\tilde{\Theta}^{(t)}}(x_s) - h_{\tilde{\Theta}'^{(t)}}(x_s)\right| + \left|h_{\tilde{\Theta}'^{(t)}}(x_s) - h_{q^{(t)}}(x_s)\right|$$

$$\leq \left(1 + \frac{4}{2\lambda_1 - 1}\right)\sqrt{\frac{2}{M}\log\left(\frac{2(T+1)^2}{\delta}\right)},$$

$$\left|h_{\tilde{\Theta}^{(t)}}(\tilde{x}) - h_{q^{(t)}}(\tilde{x})\right| \leq \left|h_{\tilde{\Theta}^{(t)}}(\tilde{x}) - h_{\tilde{\Theta}'^{(t)}}(\tilde{x})\right| + \left|h_{\tilde{\Theta}'^{(t)}}(x_s) - h_{q^{(t)}}(x_s)\right|$$

$$\leq \left(1 + \frac{4}{2\lambda_1 - 1}\right)\sqrt{\frac{2}{M}\log\left(\frac{2(T+1)^2}{\delta}\right)},$$

where we used (26). This completes the proof. $\qquad\square$

# D   Generalization Bounds for Empirical Risk Minimization

In this section, we give generalization bounds for the problem (3) in the context of *empirical risk minimization*, by using techniques developed by Chen et al. (2020). We consider the smoothed hinge loss and squared loss for binary classification and regression problems, respectively.

## D. 1 Auxiliary Results

For a set $\mathcal{F}$ of functions from a space $\mathcal{Z}$ to $\mathbb{R}$ and a set $S = \{z_i\}_{i=1}^n \subset \mathcal{Z}$, the empirical Rademacher complexity $\hat{\Re}_S(\mathcal{F})$ is defined as follows:

$$\hat{\Re}_S(\mathcal{F}) = \mathbb{E}_\sigma \left[ \sup_{f \in \mathcal{F}} \frac{1}{n} \sum_{i=1}^n \sigma_i f(z_i) \right],$$

where $\sigma = (\sigma_i)_{i=1}^n$ are i.i.d random variables taking $-1$ or $1$ with equal probability.

We introduce the uniform bound using the empirical Rademacher complexity (see Mohri et al. (2012)).

**Lemma G** (Uniform bound). *Let $\mathcal{F}$ be a set of functions from $\mathcal{Z}$ to $[-C, C]$ ($C \in \mathbb{R}$) and $\mathcal{D}$ be a distribution over $\mathcal{Z}$. Let $S = \{z_i\}_{i=1}^n \subset \mathcal{Z}$ be a set of size $n$ drawn from $\mathcal{D}$. Then, for any $\delta \in (0, 1)$, with probability at least $1 - \delta$ over the choice of $S$, we have*

$$\sup_{f \in \mathcal{F}} \left\{ \mathbb{E}_{Z \sim \mathcal{D}}[f(Z)] - \frac{1}{n} \sum_{i=1}^n f(z_i) \right\} \leq 2\hat{\Re}_S(\mathcal{F}) + 3C\sqrt{\frac{1}{2n} \log \frac{2}{\delta}}.$$

The contraction lemma (see Shalev-Shwartz and Ben-David (2014)) is useful in estimating the Rademacher complexity.

**Lemma H** (Contraction lemma). *Let $\phi_i : \mathbb{R} \to \mathbb{R}$ ($i \in \{1, \ldots, n\}$) be $\rho$-Lipschitz functions and $\mathcal{F}$ be a set of functions from $\mathcal{Z}$ to $\mathbb{R}$. Then it follows that for any $\{z_i\}_{i=1}^n \subset \mathcal{Z}$,*

$$\mathbb{E}_\sigma \left[ \sup_{f \in \mathcal{F}} \frac{1}{n} \sum_{i=1}^n \sigma_i \phi_i \circ f(z_i) \right] \leq \rho \mathbb{E}_\sigma \left[ \sup_{f \in \mathcal{F}} \frac{1}{n} \sum_{i=1}^n \sigma_i \circ f(z_i) \right].$$

Let $p_0(\theta)\mathrm{d}\theta$ be a distribution in proportion to $\exp\left(-\frac{\lambda_1}{\lambda_2} \|\theta\|_2^2\right) \mathrm{d}\theta$. We define a family of mean field neural networks as follows: for $R > 0$,

$$\mathcal{F}_{\mathrm{KL}}(R) = \{h_q : \mathcal{X} \to \mathbb{R} \mid q \in \mathcal{P}_2, \ \mathrm{KL}(q\|p_0) \leq R\}.$$

The Rademacher complexity of this function class is obtained by Chen et al. (2020).

**Lemma I** (Chen et al. (2020)). *Suppose $|h_\theta(x)| \leq 1$ holds for $\forall \theta \in \Omega$ and $\forall x \in \mathcal{X}$. We have for any constant $R \leq \frac{1}{2}$ and set $S \subset \mathcal{X}$ of size $n$,*

$$\hat{\Re}_S(\mathcal{F}_{\mathrm{KL}}(R)) \leq 2\sqrt{\frac{R}{n}}.$$

## D. 2 Generalization Bound on the Binary Classification Problems

We here give a generalization bound for the binary classification problems. Hence, we suppose $\mathcal{Y} = \{-1, 1\}$ and consider the problem (3) with the smoothed hinge loss defined below.

$$\ell(z, y) = \begin{cases} 0 & \text{if } zy \geq 1/2, \\ (1 - 2zy)^2 & \text{if } 0 \leq zy < 1/2, \\ 1 - 4zy & \text{else.} \end{cases}$$

We also define the 0-1 loss as $\ell_{01}(z, y) = \mathbb{1}[zy < 0]$.

**Theorem E.** *Let $\mathcal{D}$ be a distribution over $\mathcal{X} \times \mathcal{Y}$. Suppose there exists a true distribution $q^\circ \in \mathcal{P}_2$ satisfying $h_{q^\circ}(x)y \geq 1/2$ for $\forall(x, y) \in \mathrm{supp}(\mathcal{D})$ and $\mathrm{KL}(q^\circ\|p_0) \leq 1/2$. Let $S = \{(x_i, y_i)\}_{i=1}^n$ be training examples independently sampled from $\mathcal{D}$. Suppose $|h_\theta(x)| \leq 1$ holds for $\forall(\theta, x) \in \Omega \times \mathcal{X}$. Then, for the minimizer $q_* \in \mathcal{P}_2$ of the problem (3), it follows that with probability at least $1 - \delta$ over the choice of $S$,*

$$\mathbb{E}_{(X,Y) \sim \mathcal{D}}[\ell_{01}(h_{q_*}(X), Y)] \leq \lambda_2 \mathrm{KL}(q^\circ\|p_0) + 16\sqrt{\frac{\mathrm{KL}(q^\circ\|p_0)}{n}} + 15\sqrt{\frac{1}{2n} \log \frac{2}{\delta}}.$$

*Proof.* We first estimate a radius $R$ to satisfy $q_* \in \mathcal{F}_{\mathrm{KL}}(R)$. Note that the regularization term of objective $\mathcal{L}(q)$ is $\lambda_2 \mathrm{KL}(q\|p_0)$ and that $\ell(h_{q^\circ}(x_i), y_i) = 0$ from the assumption on $q^\circ$ and the definition of the smoothed hinge loss. Since $\mathcal{L}(q_*) \leq \mathcal{L}(q^\circ)$, we get

$$\mathrm{KL}(q_*\|p_0) \leq \frac{1}{\lambda_2}\mathcal{L}(q^\circ) = \mathrm{KL}(q^\circ\|p_0), \tag{29}$$

$$\frac{1}{n}\sum_{i=1}^{n}\ell(h_{q_*}(x_i), y_i) \leq \mathcal{L}(q^\circ) = \lambda_2\mathrm{KL}(q^\circ\|p_0). \tag{30}$$

Especially, setting $R = \mathrm{KL}(q^\circ\|p_0)$, we see $q_* \in \mathcal{F}_{\mathrm{KL}}(R)$.

We next define the set of composite functions of loss and mean field neural networks as follows:

$$\mathcal{F}(R) = \{(x,y) \in \mathcal{X} \times \mathcal{Y} \longmapsto \ell(h(x), y) \mid h \in \mathcal{F}_{\mathrm{KL}}(R)\}. \tag{31}$$

Since $\ell(z, y)$ is 4-Lipschitz continuous with respect to $z$, we can estimate the Rademacher complexity $\hat{\Re}_S(\mathcal{F}(R))$ by using Lemma H with $\phi_i(\cdot) = \ell(\cdot, y_i)$ as follows:

$$\begin{aligned}
\hat{\Re}_S(\mathcal{F}(R)) &= \mathbb{E}_\sigma\left[\sup_{h \in \mathcal{F}_{\mathrm{KL}}(R)} \frac{1}{n}\sum_{i=1}^{n}\sigma_i\ell(h(x_i), y_i)\right] \\
&\leq 4\mathbb{E}_\sigma\left[\sup_{h \in \mathcal{F}_{\mathrm{KL}}(R)} \frac{1}{n}\sum_{i=1}^{n}\sigma_i h(x_i)\right] \\
&= 4\hat{\Re}_{\{x_i\}_{i=1}^n}(\mathcal{F}_{\mathrm{KL}}(R)) \\
&\leq 8\sqrt{\frac{R}{n}}, \tag{32}
\end{aligned}$$

where we used Lemma I for the last inequality.

From the boundedness assumption on $h_q$, we have $0 \leq \ell(h_q(x), y) \leq 5$ for $\forall q \in \mathcal{P}_2$. Applying Lemma G with $\mathcal{F} = \mathcal{F}(R)$, we have with probability at least $1 - \delta$,

$$\begin{aligned}
\mathbb{E}_{(X,Y)\sim\mathcal{D}}[\ell_{01}(h_{q_*}(X), Y)] &\leq \mathbb{E}_{(X,Y)\sim\mathcal{D}}[\ell(h_{q_*}(X), Y)] \\
&\leq \frac{1}{n}\sum_{i=1}^{n}\ell(h_{q_*}(x_i), y_i) + 2\hat{\Re}_S(\mathcal{F}(R)) + 15\sqrt{\frac{1}{2n}\log\frac{2}{\delta}} \\
&\leq \lambda_2\mathrm{KL}(q^\circ\|p_0) + 16\sqrt{\frac{R}{n}} + 15\sqrt{\frac{1}{2n}\log\frac{2}{\delta}} \\
&= \lambda_2\mathrm{KL}(q^\circ\|p_0) + 16\sqrt{\frac{\mathrm{KL}(q^\circ\|p_0)}{n}} + 15\sqrt{\frac{1}{2n}\log\frac{2}{\delta}},
\end{aligned}$$

where we used $\ell_{01}(z, y) \leq \ell(z, y)$, (30) and (32). $\qquad\square$

This theorem results in the following corollary:

**Corollary C.** *Suppose the same assumptions in Theorem E hold. Moreover, we set $\lambda_1 = \lambda/\sqrt{n}$ ($\lambda > 0$) and $\lambda_2 = 1/\sqrt{n}$. Then, the following bound holds with the probability at least $1 - \delta$ over the choice of training examples,*

$$\mathbb{E}_{(X,Y)\sim\mathcal{D}}[\ell_{01}(h_{q_*}(X), Y)] \leq \frac{\mathrm{KL}(q^\circ\|p_0')}{\sqrt{n}} + 16\sqrt{\frac{\mathrm{KL}(q^\circ\|p_0')}{n}} + 15\sqrt{\frac{1}{2n}\log\frac{2}{\delta}},$$

*where $p_0'$ is the Gaussian distribution in proportion to $\exp(-\lambda\|\cdot\|_2^2)$.*

### D. 3 Generalization Bound on the Regression Problem

We here give a generalization bound for the regression problems. We consider the squared loss $\ell(z, y) = 0.5(z - y)^2$ and the bounded label $\mathcal{Y} \subset [-1, 1]$.

**Theorem F.** *Let $\mathcal{D}$ be a distribution over $\mathcal{X} \times \mathcal{Y}$. Suppose there exists a true distribution $q^\circ \in \mathcal{P}_2$ satisfying $y = h_{q^\circ}(x)$ for $\forall(x, y) \in \mathrm{supp}(\mathcal{D})$ and $\mathrm{KL}(q^\circ \| p_0) \leq 1/2$. Let $S = \{(x_i, y_i)\}_{i=1}^n$ be training examples independently sampled from $\mathcal{D}$. Suppose $|h_\theta(x)| \leq 1$ holds for $\forall(\theta, x) \in \Omega \times \mathcal{X}$. Then, for the minimizer $q_* \in \mathcal{P}_2$ of the problem ([3](#)), it follows that with probability at least $1 - \delta$ over the choice of $S$,*

$$\mathbb{E}_{(X,Y)\sim\mathcal{D}}[\ell(h_{q_*}(X), Y)] \leq \lambda_2 \mathrm{KL}(q^\circ \| p_0) + 8\sqrt{\frac{\mathrm{KL}(q^\circ \| p_0)}{n}} + 6\sqrt{\frac{1}{2n} \log \frac{2}{\delta}}.$$

*Proof.* The proof is very similar to that of Theorem [E](#). Note that $\ell(h_{q^\circ}(x_i), y_i) = 0$ from the assumption on $q^\circ$ and that inequalities ([29](#)) and ([30](#)) hold in this case too. Hence, setting $R = \mathrm{KL}(q^\circ \| p_0)$, we see $q_* \in \mathcal{F}_{\mathrm{KL}}(R)$.

Since $\ell(z, y)$ is 2-Lipschitz continuous with respect to $z \in [-1, 1]$ for any $y \in \mathcal{Y} \subset [-1, 1]$, we can estimate the Rademacher complexity $\hat{\Re}_S(\mathcal{F}(R))$ of $\mathcal{F}(R)$ (defined in ([31](#))) in the same way as Theorem [E](#):

$$\hat{\Re}_S(\mathcal{F}(R)) \leq 4\sqrt{\frac{R}{n}}. \tag{33}$$

From the boundedness assumption on $h_q$ and $\mathcal{Y}$, we have $0 \leq \ell(h_q(x), y) \leq 2$ for $\forall q \in \mathcal{P}_2$. Hence, applying Lemma [G](#) with $\mathcal{F} = \mathcal{F}(R)$, we have with probability at least $1 - \delta$,

$$\mathbb{E}_{(X,Y)\sim\mathcal{D}}[\ell(h_{q_*}(X), Y)] \leq \frac{1}{n} \sum_{i=1}^n \ell(h_{q_*}(x_i), y_i) + 2\hat{\Re}_S(\mathcal{F}(R)) + 6\sqrt{\frac{1}{2n} \log \frac{2}{\delta}}$$

$$\leq \lambda_2 \mathrm{KL}(q^\circ \| p_0) + 8\sqrt{\frac{\mathrm{KL}(q^\circ \| p_0)}{n}} + 6\sqrt{\frac{1}{2n} \log \frac{2}{\delta}},$$

where we used ([30](#)) and ([33](#)). $\qquad\square$

This theorem results in the following corollary:

**Corollary D.** *Suppose the same assumptions in Theorem [F](#) hold. Moreover, we set $\lambda_1 = \lambda/\sqrt{n}$ ($\lambda > 0$) and $\lambda_2 = 1/\sqrt{n}$. Then, the following bound holds with the probability at least $1 - \delta$ over the choice of training examples,*

$$\mathbb{E}_{(X,Y)\sim\mathcal{D}}[\ell(h_{q_*}(X), Y)] \leq \frac{\mathrm{KL}(q^\circ \| p_0')}{\sqrt{n}} + 8\sqrt{\frac{\mathrm{KL}(q^\circ \| p_0')}{n}} + 6\sqrt{\frac{1}{2n} \log \frac{2}{\delta}},$$

*where $p_0'$ is the Gaussian distribution in proportion to $\exp(-\lambda \| \cdot \|_2^2)$.*

# E Additional Discussions

## E.1 Efficient Implementation of PDA

Note that similar to SGD, Algorithm [1](#) only requires gradient queries (and additional Gaussian noise); in particular, a weighted average $\overline{g}^{(t)}$ of functions $g^{(t)}$ is updated and its derivative with respect to parameters is calculated. In the case of empirical risk minimization, this procedure can be implemented as follows. We use $\{w_i\}_{i=1}^n$ (initialized as zeros) to store the weighted sums of $\partial_z \ell(h_{\tilde{\Theta}^{(t)}}(x_{i_t}), y_{i_t})$. At step $t$ in the outer loop, $w_{i_t}$ is updated as

$$w_{i_t} \leftarrow w_{i_t} + t \partial_z \ell(h_{\tilde{\Theta}^{(t)}}(x_{i_t}), y_{i_t}).$$

The average $\nabla_{\theta_r} \overline{g}^{(t)}(\Theta^{(k)})$ can then be computed as

$$\frac{2}{\lambda_2(t+2)(t+1)} \sum_{i=1}^n w_i \partial_\theta h(\theta_r^{(k)}, x_i) + \frac{2\lambda_1 t}{\lambda_2(t+2)} \theta_r^{(k)},$$

where we use $\{\theta_r^{(k)}\}_{k=1}^M$ to denote parameters $\Theta^{(k)}$ at step $k$ of the inner loop. This formulation makes Algorithm 1 straightforward to implement.

In addition, the PDA algorithm can also be implemented with mini-batch update, in which a set of data indices $I_t = \{i_{t,1}, \ldots, i_{t,b}\} \subset \{1, 2, \ldots, n\}$ is selected per outer loop step instead of one single index $i_t$. Due to the reduced variance, mini-batch update can stabilize the algorithm and lead to faster convergence. Our theoretical results in the sequel trivially extends to the mini-batch setting.

## E. 2   Extension to Multi-class Classification

We give a natural extension of PDA method to multi-class classification settings. Let $\mathcal{C}$ denote the finite set of all class labels and $|\mathcal{C}|$ denote its cardinality. For multi-class classification problems, we define a component $h(\theta, x)$ of an ensemble as follows. Let $a_r \in \mathbb{R}^{|\mathcal{C}|}$ and $b_r \in \mathbb{R}^d$ ($r \in \{1, \ldots, M\}$) be parameters for output and input layers, respectively, and set $\theta_r = (a_r, b_r)$ and $\Theta = \{\theta_r\}_{r=1}^M$. Then, we define $h_{\theta_r}(x) = h(\theta, x) = \sigma_2(a_r \sigma_1(b_r^\top x))$[7] which is a neural network with one hidden neuron, and denote

$$h_\Theta(x) = \frac{1}{M} \sum_{r=1}^M h_{\theta_r}(x).$$

Note that $h_\Theta(x)$ is a natural two-layer neural network with multiple outputs. Suppose that each parameter $\theta_r$ follows $q(\theta)\mathrm{d}\theta$. Then the mean field limit can be defined as

$$h_q(\cdot) = \mathbb{E}_{\theta \sim q}[h_\theta(\cdot)] : \mathbb{R}^d \to \mathbb{R}^{|\mathcal{C}|}.$$

Let $\ell(z, y)$ ($z = \{z_y\}_{y \in \mathcal{C}} \in \mathbb{R}^{|\mathcal{C}|}, y \in \mathcal{C}$) be the loss for multi-class classification problems. A typical choice is the cross-entropy loss with the soft-max activation, that is

$$\ell(z, y) = -\log \frac{\exp(z_y)}{\sum_{y' \in \mathcal{C}} \exp(z_{y'})} = -z_y + \log \sum_{y' \in \mathcal{C}} \exp(z_{y'}).$$

In this case, the functional derivative of $\ell(h_q(x), y)$ with respect to $q$ is

$$-h_y(\theta, x) + \frac{\sum_{y' \in \mathcal{C}} \exp(h_{q,y'}(x)) h_{y'}(\theta, x)}{\sum_{y' \in \mathcal{C}} \exp(h_{q,y'}(x))}$$

where we supposed the outputs of $h_\theta$ and $h_q$ are also indexed by $\mathcal{C}$. Hence, the counterpart of $g^{(t)}$ in Algorithm 2 in this setting is

$$g^{(t)} = -h_{y_t}(\cdot, x_t) + \frac{\sum_{y' \in \mathcal{C}} \exp(h_{q^{(t)},y'}(x_t)) h_{y'}(\cdot, x_t)}{\sum_{y' \in \mathcal{C}} \exp(h_{q^{(t)},y'}(x_t))} + \lambda_1 \| \cdot \|_2^2.$$

Using this function, the DA method for multi-class classification problems can be obtained in the same manner as Algorithm 2. Moreover, its discretization can be also immediately derived by replacing the function $\overline{g}^{(t)}$ used in Algorithm 1 with

$$\overline{g}^{(t)} = \frac{2}{\lambda_2(t+2)(t+1)} \sum_{s=1}^t s \left( -h_{y_s}(\cdot, x_s) + \frac{\sum_{y' \in \mathcal{C}} \exp(h_{\tilde{\Theta}^{(s)},y'}(x_s)) h_{y'}(\cdot, x_s)}{\sum_{y' \in \mathcal{C}} \exp(h_{\tilde{\Theta}^{(s)},y'}(x_s))} + \lambda_1 \| \cdot \|_2^2 \right).$$

In the case of empirical risk minimization, we can adopt an efficient implementation as done in Section E. 1. We use $\{w_{i,y}\}_{i \in \{1,\ldots,n\}, y \in \mathcal{C}}$ (initialized as zeros) to store the coefficients of $h_y(\cdot, x_i)$. At step $t$ in the outer loop, $w_{i_t,y}$ ($y \in \mathcal{C}$) are updated as

$$w_{i_t,y} \leftarrow \begin{cases} w_{i_t,y} + t \left( -1 + \dfrac{\exp(h_{\tilde{\Theta}^{(t)},y}(x_{i_t}))}{\sum_{y' \in \mathcal{C}} \exp(h_{\tilde{\Theta}^{(t)},y'}(x_{i_t}))} \right) & y = y_{i_t}, \\[4mm] w_{i_t,y} + t \dfrac{\exp(h_{\tilde{\Theta}^{(t)},y}(x_{i_t}))}{\sum_{y' \in \mathcal{C}} \exp(h_{\tilde{\Theta}^{(t)},y'}(x_{i_t}))} & y \neq y_{i_t}. \end{cases}$$

---

[7]Here, $a_r \sigma_1(b_r^\top x)$ is a scalar $\sigma_1(b_r^\top x)$ times a vector $a_r$.

Then, $\nabla_{\theta_r} \bar{g}^{(t)}(\Theta^{(k)})$ can be computed as

$$\frac{2}{\lambda_2(t+2)(t+1)} \sum_{i=1}^{n} \sum_{y \in \mathcal{C}} w_{i,y} \partial_\theta h_y(\theta_r^{(k)}, x_i) + \frac{2\lambda_1 t}{\lambda_2(t+2)} \theta_r^{(k)},$$

where we use $\{\theta_r^{(k)}\}_{k=1}^{M}$ to denote parameters $\Theta^{(k)}$ at step $k$ of the inner loop.

Finally, we remark that while we here utilize a simple network $h_\theta(x)$ to recover a normal two-layer neural network, it is also possible to use deep narrow networks or narrow convolutional neural networks as a component $h_\theta(x)$; in other words $h_\Theta$ can represent an ensemble of various types of small network. While such extensions are not covered by our current theoretical analysis, they may achieve better practical performance.

### E. 3   Correspondence with Finite-dimensional Dual Averaging Method

We explain the correspondence between the finite-dimensional dual averaging method developed by Nesterov (2005, 2009); Xiao (2009) and our proposed method (Algorithm 2); our goal here is to provide an intuitive understanding of the derivation of Algorithm 2 in the context of the classical dual averaging method.

First, we introduce the (regularized) dual averaging method (Nesterov, 2009; Xiao, 2009) in a more general form for solving the regularized optimization problem on the finite-dimensional space. Let $w \in \mathbb{R}^m$ be a parameter, $l(w, z)$ be a convex loss in $w$, where $z$ is a random variable which represents an example, and $\Psi(w)$ is a regularization function. Then, the problem solved by the dual averaging method is given as

$$\min_{w \in \mathbb{R}^m} \left\{ \mathbb{E}_z[l(w, z)] + \Psi(w) \right\}.$$

Let $\{w^{(s)}\}_{s=1}^{t}$ and $\{f^{(s)}\}_{s=1}^{t} = \{\partial_w l(w^{(s)}, z_s)\}_{s=1}^{t}$ be histories of iterates and stochastic gradients. The subproblems to produce the next iterate in the dual averaging method is designed by using the strongly convex function $d(w)$ and positive hyperparameters $\{\alpha_s\}_{s=1}^{\infty}$ and $\{\beta_s\}_{s=2}^{\infty}$. Specifically, the next iterate $w^{(t+1)}$ is defined as the minimizer of the following problem in which the loss function is linearized and weighted sum of which is taken over the history:

$$\min_{w \in \mathbb{R}^m} \left\{ \sum_{s=1}^{t} \alpha_s f^{(s)\top} w + \sum_{s=1}^{t} \alpha_s \Psi(w) + \beta_{t+1} d(w) \right\}. \tag{34}$$

Next, we consider our problem setting of optimizing the probability distribution and reformulate the subproblem (7) solved in Algorithm 2 as follows:

$$\min_{q \in \mathcal{P}_2} \left\{ \mathbb{E}_q \left[ \sum_{s=1}^{t} s g^{(s)} \right] + \sum_{s=1}^{t} s \lambda_2 \mathbb{E}_q[\log(q)] + (t+1)\lambda_2 \mathbb{E}_q[\log(q)] \right\}, \tag{35}$$

By comparing (34) and (35), we arrive at the following correspondence: $\alpha_s = \beta_s = s$, $f^{(s)} \sim g^{(s)}$, $d(w) = \Psi(w) \sim \lambda_2 \mathbb{E}_q[\log(q)]$. We note that in our problem setting the expectation by $q$ can be seen as an inner product with the integrand and $\lambda_2 \mathbb{E}_q[\log(q)]$ is also set to $d(w)$ because *the negative entropy acts as a strongly convex function* (Lemma A).

## F   Additional Experiments

### F. 1   Comparison of Generalization Error

We provide additional experimental results on the generalization performance of PDA. We consider empirical risk minimization for a regression problem (squared loss): the input $x_i \sim \mathcal{N}(0, I_p)$, and $f_*$ is a single index model: $f_*(x) = \text{sign}(\langle w_*, \mathbf{x} \rangle)$. W set $n = 1000$, $p = 50$, $M = 200$, and implement both noisy gradient descent (Mei et al., 2018) using full-batch gradient and our proposed Algorithm 1 (PDA) using mini-batch update with batch size 50.

Figure 3 we compare the generalization performance of different training methods: noisy GD and PDA in the mean field regime, and also noisy GD in the kernel regime. We fix the $\ell_2$ and entropy regularization to be the same across all settings: $\lambda_1 = 10^{-2}$, $\lambda_2 = 5 \times 10^{-4}$. We set the *total* number of iterations (outer + inner loop steps) in PDA to be the same as GD, and tuned the learning rate for optimal generalization. Observe that

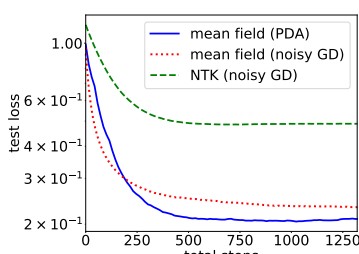

Figure 3: Test error of mean field neural networks ($\alpha = 1$) trained with noisy GD (red) and PDA (blue), and network in the kernel regime ($\alpha = 1/2$) optimized by GD (green).

- Model with the NTK scaling (green) generalizes worse than the mean field models (red and blue). This is consistent with observations in Chizat and Bach (2018a).

- For the mean field scaling, PDA (under early stopping) leads to slightly lower test error than noisy GD. We intend to further investigate this difference in the generalization performance. (see Appendix D for generalization bounds of the PDA solution)

### F. 2  PDA Beyond $\ell_2$ Regularization

Note that our current formulation (4) considers $\ell_2$ regularization, which allows us to establish polynomial runtime guarantee for the inner loop via the Log-Sobolev inequality. As remarked in Section 4, our global convergence analysis can easily be extended to Hölder-smooth gradient via the convergence rate of Langevin algorithm given in Erdogdu and Hosseinzadeh (2020). Although we do not provide details for this extension in the current work (due to the use of Vempala and Wibisono (2019)), we empirically demonstrate one of its applications in handling $\ell_p$ regularized objectives for $p > 1$ in the following form,

$$R^p_{\lambda_1, \lambda_2}(q) \overset{\text{def}}{=} \lambda_1 \mathbb{E}_q[\|\theta\|_p^p] + \lambda_2 \mathbb{E}_q[\log(q)]. \tag{36}$$

Erdogdu and Hosseinzadeh (2020) cannot directly cover the non-smooth $\ell_1$ regularization, but we can still obtain relatively sparse solution by setting $p$ close to 1. Intuitively speaking, when the underlying task exhibits certain low-dimensional or sparse structure, we expect a sparsity-promoting regularization to achieve better generalization performance.

Figure 4(a) demonstrates the advantage of $L_p$-norm regularization for $p < 2$ in empirical risk minimization, when the target function exhibits sparse structure. We set $n = 1000, p = 50$; the teacher is a multiple-index model ($m = 2$) with binary activation, and parameters of each neuron are 1-sparse. We optimize the student model with PDA (warm-start), where we set $\lambda_1 = 10^{-2}$, $\lambda_2 = 10^{-4}$, and vary the norm penalty $p$ from 1.01 to 2. Note that smaller $p$ results in favorable generalization due to the induced sparsity. On the other hand, we expect the benefit of sparse regularization to diminish when the target function is not sparse. This intuition is confirmed in 4(b), where we control the target sparsity by randomly selecting $r$ parameters to be non-zero, and we define $s = r/d$ to be the sparsity level. Observe that the benefit of sparsity-inducing regularization (smaller $p$) is more prominent under small $s$ (brighter color), which indicates a sparse target function.

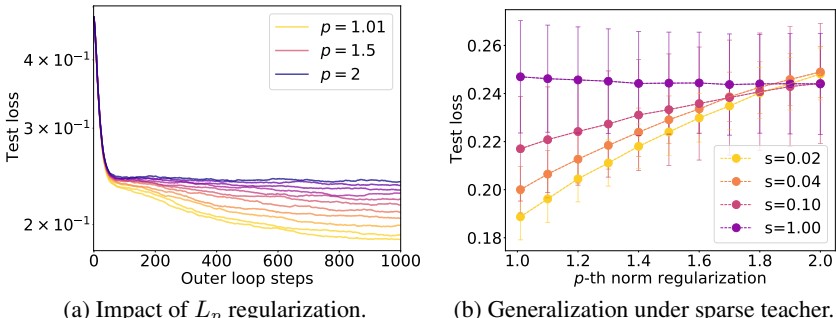

(a) Impact of $L_p$ regularization.          (b) Generalization under sparse teacher.

Figure 4: PDA with general $\ell_p$ regularizer (objective (36)). (a) Generalization error vs. training time in learning a 1-sparse target function. (b) generalization error vs. sparsity of the target function $s$.

## F. 3 On the Role of Entropy Regularization

Our objective (3) includes an entropy regularization with magnitude $\lambda_2$. In this section we illustrate the impact of this regularization term. In Figure 5(a) we consider a synthetic 1D dataset ($n = 15$) and plot the output of a two-layer tanh network with 200 neurons trained by SGD and PDA to minimize the *squared loss* till convergence. We use the same $\ell_2$ regularization ($\lambda_1 = 10^{-3}$) for both algorithms, and for PDA we set the entropic term $\lambda_2 = 10^{-4}$. Observe that SGD with weak regularization (red) almost interpolates the noisy training data, whereas PDA with entropy regularization finds low-complexity solution that is smoother (blue).

We therefore expect entropy regularization to be beneficial when the labels are noisy and the underlying target function (teacher) is "simple". We verify this intuition in Figure 5(b). We set $n = 500$, $d = 50$ and $M = 500$, and the teacher model is a linear function on the input features. We employ SGD or PDA to optimize the squared error. For both algorithms we use the same $\ell_2$ regularization $\lambda_1 = 10^{-2}$, but PDA includes a small entropy term $\lambda_2 = 5 \times 10^{-4}$. We plot the generalization error of the converged model under varying amount of label noise. Note that as the labels becomes more corrupted, PDA (blue) results in lower test error due to the entropy regularization[8]. On the other hand, model under the kernel scaling (green) generalizes poorly compared to the mean field models. Furthermore, Figure 5(c) demonstrates that entropy regularization can be beneficial under low noise (or even noiseless) cases as well. We construct the teacher model to be a multiple-index model with binary activation. Note that in this setting PDA achieves lower stationary risk across all noise level, and the advantage amplifies as labels are further corrupted.

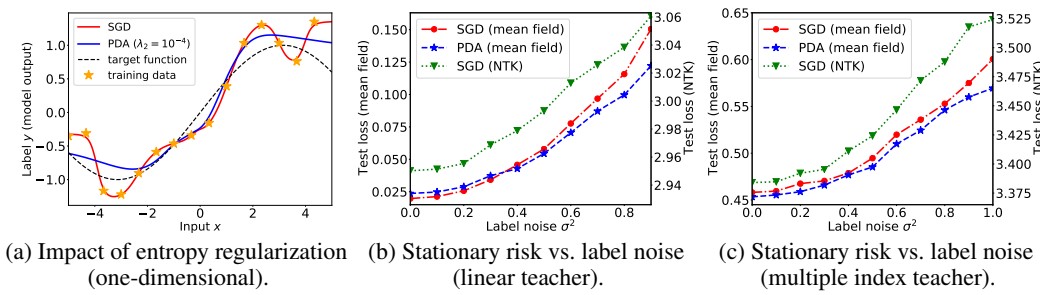

| (a) Impact of entropy regularization (one-dimensional). | (b) Stationary risk vs. label noise (linear teacher). | (c) Stationary risk vs. label noise (multiple index teacher). |

Figure 5: (a) 1D illustration of the impact of entropy regularization in two-layer tanh network: PDA (blue) finds a smoother solution that does not interpolate the training data due to entropy regularization. (b)(c) Test error of two-layer tanh network trained till convergence. PDA (blue) becomes advantageous compared to SGD (red) when labels become noisy, and the NTK model (green, note that the y-axis is on different scale) generalizes considerably worse than the mean field models.

## F. 4 Adaptivity of Mean Field Neural Networks

Recall that one motivation to study the mean field regime (instead of the kernel regime) is the presence of *feature learning*. We illustrate this behavior in a simple student-teacher setup, where the target function is a single-index model with tanh activation. We set $n = 500, d = 50$, and optimize a two-layer tanh network ($M = 1000$), either in the mean field regime using PDA, or in the kernel regime using SGD. For both methods we choose $\lambda_1 = 10^{-3}$, and for PDA we choose $\lambda_2 = 10^{-4}$.

In Figure 6 we plot the the evolution of the cosine similarity between the target vector $w^*$ and the top-5 singular vectors (PC1-5) of the weight matrix during training. In Figure 6(a) we observe that the mean field model trained with PDA "adapts" to the low-dimensional structure of the target function; in particular, the leading singular vector (bright yellow) aligns with the target direction. In contrast, we do not observe such alignment on the network in the kernel regime (Figure 6(b)), because the parameters do not travel away from the initialization. This comparison demonstrates the benefit of the mean field parameterization.

---

[8]Note that entropy regularization is not the only way to reduce overfitting – such capacity control can also be achieved by proper early stopping or other types of explicit regularization.

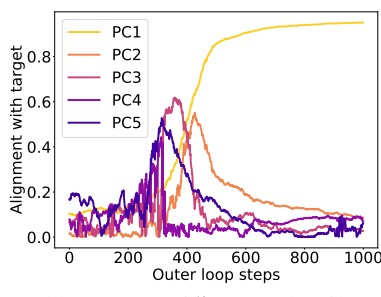 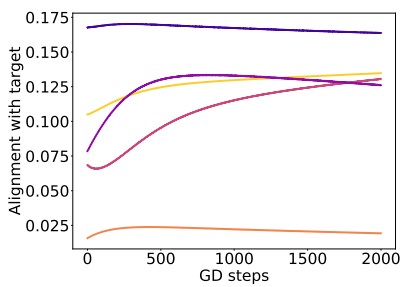

(a) Parameter Alignment (PDA).  (b) Parameter Alignment (NTK).

Figure 6: Cosine similarity between the target vector $w^*$ and the top-5 singular vectors (PC1-5) of the weight matrix during training. The learned parameters "align" with the target function under the mean field parameterization (a), but not the NTK parameterization (b).

# G  Additional Related Work

**Particle inference algorithms.**  Bayesian inference is another example distribution optimization, in which the objective is to minimize an entropic regularized linear functional. In addition to the Langevin algorithm, several interacting particle methods have been developed for this purpose, such as particle mirror descent (PMD) (Dai et al., 2016), Stein variational gradient descent (SVGD) (Liu and Wang, 2016), and ensemble Kalman sampler (Garbuno-Inigo et al., 2020), and the corresponding mean field limits have been analyzed in Lu et al. (2019); Ding and Li (2019). We remark that naive gradient-based method on the probability space often involves computing the probability of particles for the entropy term (e.g., kernel density estimation in PMD), which presents significant difficulty in constructing particle inference algorithms. In contrast, our proposed algorithm avoids this computational challenge due to its algorithmic structure.

**Optimization of probability distributions.**  Parallel to our work, several recent papers also proposed optimization methods over space of probability measures by adapting finite-dimensional convex optimization theory. Ying (2020), Kent et al. (2021) and Chizat (2021) extend the Mirror descent method, Frank-Wolfe method, and (accelerated) Bregman proximal gradient method to the optimization of probability measures, respectively. In addition, Hsieh et al. (2019) developed an entropic mirror descent algorithm for generative adversarial networks, and Chu et al. (2019) analyzed probability functional descent in the context of variational inference and reinforcement learning.

**The kernel regime and beyond.**  The neural tangent kernel model (Jacot et al., 2018) describes the learning dynamics of neural network under appropriate scaling. Such description builds upon the linearization of the learning dynamics around its initialization, and (quantitative) global convergence guarantees of gradient-based methods for neural networks can be shown for regression problems (Du et al., 2019; Allen-Zhu et al., 2019; Zou et al., 2020; Nitanda and Suzuki, 2020) as well as classification problems (Cao and Gu, 2019; Nitanda et al., 2019; Ji and Telgarsky, 2019).

However, due to the linearization, the NTK model cannot explain the presence of "feature learning" in neural networks (i.e. parameters are able to travel and adapt to the structure of the learning problem). In fact, various works have shown that deep learning is more powerful than kernel methods in terms of approximation and estimation error (Suzuki, 2018; Ghorbani et al., 2019b; Suzuki and Nitanda, 2019; Schmidt-Hieber, 2020; Ghorbani et al., 2020; Imaizumi and Fukumizu, 2020), and in certain settings, neural networks optimized with gradient-based methods can outperform the NTK model (or more generally any kernel methods) in terms of generalization error or excess risk (Allen-Zhu and Li, 2019; Ghorbani et al., 2019a; Yehudai and Shamir, 2019; Bai and Lee, 2019; Allen-Zhu and Li, 2020; Li et al., 2020; Suzuki, 2020; Daniely and Malach, 2020).