# OpenReview forum: "Particle Dual Averaging: Optimization of Mean Field Neural Network with Global Convergence Rate Analysis"
_NeurIPS.cc/2021/Conference — NeurIPS 2021 Poster_

### Official Review · Reviewer_fmps · 2021-07-15

**Rating:** 7
**Confidence:** 4

**Summary:**

This paper introduces a Particle Dual Averaging (PDA) algorithm to minimize non-linear functional of probability distributions. One of the main application of this algorithm is the training of two-layer neural networks. The authors obtain quantitative convergence results for PDA under mild assumption on the neural network for the minimization of an entropy-regularized loss function. In particular they show that a solution can be reached with precision $O(\varepsilon)$ after $O(\varepsilon^{-3})$ iterations. The theoretical study is completed with the investigation of the finite particle algorithm and generalization bounds. The authors then present a short experimental study on regression and classification tasks, illustrating the properties of the algorithm.

**Limitations And Societal Impact:**

Societal impact is irrelevant for this very theoretical work. The limitations of the algorithm could have been better highlighted and addressed (increasing cost of the inner loop, marginal performance over SGD), see my previous comments.

**Main Review:**

STRENGTHS:
First of all the paper is very well-written. The motivation and the algorithm are clearly presented and the theoretical results are given with precise assumptions.

The idea of using dual averaging for probabilistic optimization is original and really promising. One of the key remark that minimizers of linearized functionals on probability measures with Kullback-Leibler regularization are given by Gibbs measures might find applications outside of the context of overparameterized neural networks.

The authors use recent tools in optimization/sampling (convergence in Kullback-Leibler divergence of the Langevin algorithm under log-Sobolev condition with quantitative rates, convergence of the dual averaging algorithm) in order to derive quantitative results. These quantitative results are really promising.

WEAKNESSES:
One of the main weakness of the paper is its experiments which I found to be a bit too limited even for a theoretical paper. The authors could have dealt with real datasets for classification such as MNIST or CIFAR-10 in order to assess the scalability of their method with respect to the dimension and hence its applicability.

Similarly, even if some comparison with a classical training of neural networks (with SGD) are reported in the appendix, I think further comparisons should be conducted to assess the efficiency of the proposed methodology (for instance in Figure 4 in the appendix, PDA seems to perform only marginally better than SGD which is simpler to train). Also instead of SGD the author might consider sampling with a Langevin algorithm (i.e. changing the noise scaling). Doing so incorporates an entropic term and will allow for a fairer comparison with PDA in my opinion.

From a theoretical point of view I think that the authors should discuss in greater details the relationship between their algorithm and other works. In particular, how does the proposed approach relates to the ones of [1,2] which prove theoretical results for either SGD or the Langevin dynamics and not the PDA algorithm.

[1] Hu, Ren, Siska, Szpruch - Mean-field Langevin dynamics and energy landscape of neural networks
[2] Javanmard, Mondelli, Montanari - Analysis of a two-layer neural network via displacement convexity

COMMENTS:
-The original dual averaging algorithm allows flexibility in the primal and dual discretization steps, see (2.14) in [3]. In this work the authors have chosen $\beta_k = O(k^2)$ and $\lambda_k = O(k)$ if I am not mistaken. This choice of parameters is not discussed in the paper.
-Related to this remark, one of the limitation of the algorithm is that each inner-loop gets more and more expensive. Is this due to the way the parameters of the dual averaging are set? Is there any way to mitigate this issue?

MINOR COMMENTS:
-I am not sure to understand the role of $n$ in Proposition D. I assume that $x_i$ are the generated the test data points?

[3] Nesterov - Primal-dual subgradient methods for convex problems

**Time Spent Reviewing:**

20

---

> ### Author Response · Authors · 2021-08-10
> **Response to Reviewer fmps**
>
> Thank you for thorough reading and helpful comments. The technical comments are addressed below.
>
> - **"how does the proposed approach relates to the ones of [1,2] which prove theoretical results for either SGD or the Langevin dynamics and not the PDA algorithm."**
>
> [1] Hu, Ren, Siska, Szpruch. *Mean-field Langevin dynamics and energy landscape of neural networks.*
>
> [2] Javanmard, Mondelli, Montanari. *Analysis of a two-layer neural network via displacement convexity.*
>
> The major difference is the convergence of PDA is guaranteed for wider range of problems than [1,2].
> For instance, the loss function in [1] is limited to the type $\Phi( y - f(x) )$ where $\Phi$ is a convex function and $\Phi \geq 0, \Phi(0)=0$, and the [2] focuses on the squared loss and makes a strong assumption on the target function to be strongly concave.
> On the other hand, we do not rely on such assumptions on the target function and loss functions; indeed our theory covers general loss functions including the squared loss and logistic loss functions. Moreover, our convergence rate directly applies to the discrete-time and finite-particle setting, and our convergence rate holds for any choice of regularization parameters.
> For other theoretical differences, see comments to Reviewer kvcd.
>
> - **"One of the main weakness of the paper is its experiments which I found to be a bit too limited"**
>
> We acknowledge that our experimental results are not extensive. The primary focus of our work is to theoretically demonstrate that two-layer neural networks in the mean field regime can be efficiently optimized with convergence rate guarantees. Therefore we only performed simple experiments to support our theoretical results (similar to [Chizat and Bach 2018],[Mei et al. 2018]).
>
> We also agree that the comparison between SGD and PDA can be refined by including the entropy term for SGD — we will take this into account in the revision.
> It is worth noting that we do not claim that PDA is always more efficient than SGD: while PDA is guaranteed to converge at a $1/T$ rate as specified in Theorem 1 and empirically confirmed in Figure 2(a), it is possible that in some specific settings, SGD also converges globally at an even faster rate (that has not been established theoretically).
> Understanding when one algorithm outperforms the other in terms of convergence rate is an interesting future direction.
>
>
> - **"The original dual averaging algorithm allows flexibility in the primal and dual discretization steps, see (2.14) in [3]. In this work the authors have chosen $\beta_k=O(k^2)$ and $\lambda_k=O(k)$ if I am not mistaken."**
>
> [3] Nesterov. *Primal-dual subgradient methods for convex problems.*
>
> As the reviewer correctly pointed out, PDA can be instantiated from [3] by setting $\beta_k=O(k^2)$ and $\lambda_k=O(k)$. However, since we consider the regularized problem, part of $\beta_k$ should be interpreted as a coefficient of the regularized objective.
> That is, we also use the entropy term as a prox-function in dual averaging, leading to a slightly larger coefficient of the negative entropy than that of loss function. Specifically, a sub-problem solved in the method is given as
>
> $$E_q \left[ \sum_{s=1}^t sg^{(s)} \right] + \frac{\lambda_2}{2}(t+2)(t+1)E_q[\log(q)],$$
>
> and in our theory this can be interpreted as
>
> $$ E_q\left[ \sum_{s=1}^t s g^{(s)}  \right] + \left(\sum_{s=1}^t s\right) \lambda_2 E_q[\log(q)] + (t+1)\lambda_2 E_q[\log(q)]. $$
>
> Hence, $\beta_k=O(k)$ and $\lambda_k=O(k)$ is an appropriate interpretation in regularized problems and we need to extend the original dual averaging [3] to the regularized dual averaging as done in [L. Xiao (2009)] (in which $\lambda_k$ is set to a constant, though).
> A benefit of our hyperparameter setting is an improvement of logarithmic-factor appearing in [L. Xiao (2009)].
> We will add the above discussion to the Appendix of the revised manuscript.
>
>
> - **"one of the limitation of the algorithm is that each inner-loop gets more and more expensive. Is this due to the way the parameters of the dual averaging are set? Is there any way to mitigate this issue?"**
>
> Indeed due to the required precision for the outer-loop convergence, the number of inner-loop steps need to increase with time.
> This being said, we remark that since $\partial_z \ell( h_{\tilde{\Theta}^{(s)}}(x_s),y_s)$ are constants among all iterations, it is enough to compute these values only once. Moreover, for empirical risk minimization we can run the algorithm with the same per-iteration computational cost as gradient descent by the efficient implementation (see Section F).
>
> - **"I am not sure to understand the role of $n$ in Proposition D. I assume that
> $x_i$ are the generated the test data points?''**
>
> Thank you for pointing this out. We here considered empirical risk minimization and so $x_i$ represents the training example.
> But this result is not limited to training examples in empirical risk minimization, and holds any data examples -- we will update the notation to include the prediction on training examples in the online learning setting and on test examples.

---

> > ### Comment · Reviewer_fmps · 2021-08-23
> > **Thank you for your response**
> >
> > Thank you for your response and your clarifications.
> >
> > I agree that experimental results are not the primary concerns of this study as in [Chizat and Bach 2018],[Mei et al. 2018].
> > However, in these two papers the studied methods correspond to a continuous-time version of the gradient descent algorithm (or the Langevin algorithm in one section of [Mei et al. 2018]) whose stochastic versions are widely used in ML.
> > The algorithm introduced here (although it encompasses the one of [Mei et al. 2018] in a specific case) has not been used in challenging experimental settings yet.
> >
> > I thank the authors for their interesting clarifications on the links with [Nesterov, Primal-dual subgradient methods for convex problems].
> > I think that such a discussion should definitely be included in the paper.
> >
> > Also, I am not really sure to understand the author response to my comment regarding the fact that the number of inner-loop steps need to increase with time. I don't really see how the implementation of Section F mitigates this issue.

---

> > > ### Author Response · Authors · 2021-08-25
> > > **Thank you for the followup.**
> > >
> > > Thank you for the helpful additional comments. We will include more discussion on the link with primal-dual subgradient methods.
> > > We make two additional remarks.
> > >
> > > - We agree that our proposed method has not been applied to challenging experimental settings such as large-scale image classification; characterizing the effectiveness of our approach in real-world problems is an interesting future direction.
> > > We also note that the two-layer MLP considered in our work (and many other papers on the mean field regime) is an idealized and simplified representation of practical neural networks. Our motivation is that even in this simplified setting, establishing quantitative convergence rate guarantee beyond the kernel regime (i.e., in the presence of feature learning) is challenging. We believe that extending this theoretical analysis to more practical settings such as deeper networks is an important research problem.
> > >
> > > - Regarding the increasing inner loop complexity, note that if the required optimization accuracy $\epsilon$ is predefined before we run the algorithm, then we can use a constant number of inner loop steps depending on this accuracy. But as the required accuracy becomes smaller, the number of inner loop steps should be increased.
> > > This is because the subproblems in the original dual averaging method are supposed to be solved exactly for convergence. Hence, the optimization gaps of these subproblems are expected to become smaller as the outer loop proceeds.
> > > We remark that this increased inner loop complexity is rather common for an inexact variant of proximal-type methods whose solutions of subproblems cannot be explicitly computed (because inexactness for subproblems basically yields an inexact solution for the original problem), such as the following paper.
> > > Mark Schmidt, Nicolas Le Roux, and Francis Bach. *Convergence Rates of Inexact Proximal-Gradient Methods for Convex Optimization*.

---

> > > > ### Comment · Reviewer_gSde · 2021-08-31
> > > > **On the computational complexity of PDA**
> > > >
> > > > I would like to second Reviewer fmps's concern about computational complexity.
> > > > My understanding is that the efficient implementation described in Appendix F reduces the complexity of one inner loop iteration from O(tD) to O(nD), where t is the step number, n is the number of training examples, and D is the total dimension of the neural network weights. While it's true that this is an improvement compared to O(tD) when t >> n, this is still too expensive, one inner loop iteration has the same cost as a *full* gradient evaluation, which is much more expensive than SGD.
> > > > Can the authors clarify if this is indeed the case, and whether this can be further mitigated? Thank you.

---

> > > > > ### Author Response · Authors · 2021-09-01
> > > > > **Per-iteration computational cost.**
> > > > >
> > > > > Thank you for the followup. As you pointed out, the per-iteration cost of our method scales with the number of training examples and is the same as normal gradient descent.
> > > > > This is why we commented on lines 179-185 that PDA can be seen as a slight modification of gradient descent; in particular, the algorithmic difference between the two is just the coefficients of $\partial_\theta h(\theta,x_i)$ (see lines 905-906 in Appendix). While the per-iteration computational cost of PDA is larger than SGD, we believe this does not undermine our theoretical contribution, because global convergence analysis of deterministic optimization methods (such as gradient descent) for mean field neural network is already challenging, and indeed many related studies ([Chizat and Bach (2018)], [Hu et al. (2019)], etc.) focus on deterministic gradient descent rather than the stochastic variants; we also note that under similar setting, the total complexity of SGD to converge to the global solution is theoretically unknown (and possibly slower than PDA) due to the lack of convergence rate analysis.
> > > > >
> > > > > As for ways to reduce the per-iteration cost, since the potential function of the intermediate Gibbs distribution can be written as a finite-sum, *stochastic gradient* Langevin Monte Carlo and its variants may be used to replace our (full gradient) Langevin algorithm, though additional effort is required to rigorously justify this replacement.
> > > > > We thank the reviewer for suggesting this important future extension.

---

> > > > > > ### Comment · Reviewer_gSde · 2021-09-01
> > > > > > **Re: Per-iteration computational cost**
> > > > > >
> > > > > > Thanks for confirming the per-iteration cost. I agree that this does not detract from the theoretical contribution of the paper.
> > > > > > The paragraph line 179 does indeed mention that this is closely related to gradient descent, thank you for pointing that out. The last sentence in that paragraph might be ambiguous: it says that the algorithm can be extended to the minibatch variant; for efficient implementation ... see Appendix F. This could be interpreted to mean efficient implementation of a minibatch gradient version, it would be good to remove this ambiguity in the revision. Also please clarify the exact setting in experiment G.1, in particular what is meant by "PDA using mini-batch update": does this mean mini-batching in the outer loop, or when computing the gradient in the Langevin step, or both? If only in the outer loop, then the per-iteration cost of PDA and SGD in Figure 3 are not comparable, and the reader should be made aware of this. Thanks.

---

> > > > > > > ### Author Response · Authors · 2021-09-02
> > > > > > > **Mini-batch update.**
> > > > > > >
> > > > > > > Thank you for the close reading. Indeed the mini-batch version of PDA used in our experiments is only for the outer loop, and the inner loop utilizes the full gradient Langevin algorithm. Thus as you pointed out, the per-iteration costs of PDA and SGD in Appendix G.1 Figure 3 are not comparable (we however note that due to commonly-used parallelization, the wall time of running the PDA inner loop does not scale linearly with the batch size). Following your suggestion, we will revise our description of the mini-batch variant.
> > > > > > >
> > > > > > > In addition, we ran the comparison between PDA and full-batch gradient descent (which has similar per-iteration complexity) in the same setting as Figure 3, and observed that the trend is qualitatively similar to Figure 3. We will add this figure to Section G.1 for the sake of fair comparison and clarify the experimental setting.

---

### Official Review · Reviewer_QjGu · 2021-07-16

**Rating:** 7
**Confidence:** 3

**Summary:**

In this paper the authors  propose the particle dual averaging (PDA) method, which generalizes the dual averaging method in convex optimization to the optimization over probability distributions with quantitative runtime guarantee. The method in thi paper can thus be interpreted as an extension of the Langevin algorithm to naturally handle nonlinear functional on the probability space. An important application of the proposed method is the optimization of neural network in the mean field regime. By adapting finite-dimensional convex optimization theory into the space of distributions, the athors  establish global convergence of PDA for two-layer mean field neural networks under more general settings and simpler analysis. Numerics support the   theoretical findings.

**Limitations And Societal Impact:**

No.

**Main Review:**

This paper  develops optimization algorithms called PDA for neural networks in the mean field regime with
accurate quantitative guarantees as  the kernel regime enjoys,  which is an important issue in understanding the
optimization error in deep learnling.  Althrouth the ideas and techniques used in this paper are not new, the authors
establish quantitative global convergence rate of PDA in minimizing an KL-regularized objective  for the two layer neural network opitmization.   Overall,  this paper gives a nice contribution in understanding optimization in deep learning.

One question: Does the constant hidden in $\tilde{O}(\epsilon^3)$ depend on the ambient dimension? If so, what is the quantitive
relation between them?

**Time Spent Reviewing:**

5

---

> ### Author Response · Authors · 2021-08-10
> **Response to Reviewer QjGu**
>
> Thank you for the positive feedback. We address the technical comment below.
>
> - **"Does the constant hidden in $\tilde{\mathcal{O}}(\epsilon^3)$ depend on the ambient dimension? If so, what is the quantitative relation between them?"**
>
> Since the total runtime complexity is obtained by the product of the outer loop complexity and maximal inner loop complexity (see also reply to Reviewer kvcd), we can see that dependency on the dimensionality $p$ in our $\tilde{\mathcal{O}}(\epsilon^3)$ rate is $p^2$ by combining Theorem 1 and Corollary 2.

---

> > ### Comment · Reviewer_QjGu · 2021-08-23
> > **Thanks the authors' reply.**
> >
> >  Thanks the authors' reply. My concerns are elimnated.

---

> > > ### Author Response · Authors · 2021-08-25
> > > **Thank you for the followup.**
> > >
> > > We are delighted to see that our reply addressed your concerns.
> > > We would like to make an additional remark on the novelty of our proposed method (in response to the comment that *"the ideas and techniques used in this paper are not new"* in the original review):
> > > While dual averaging is a well-known technique in convex optimization, adapting this method to our problem setting (optimization in the space of measures) is technically nontrivial and novel to our knowledge. Moreover, this allows us to establish quantitative global convergence guarantee (for the discrete-time algorithm) over a wider range of regularization choices than prior analysis on the mean field Langevin dynamics -- please see reply to Reviewer kvcd for details.
> > >
> > > We would appreciate if you could update your evaluation in light of our response.
> > > We would be happy to answer any further questions in the discussion period.

---

### Official Review · Reviewer_gSde · 2021-07-16

**Rating:** 7
**Confidence:** 4

**Summary:**

The paper is motivated by optimizing two-layer neural networks, with entropy and quadratic regularization. The proposed method is an extension of the dual averaging method to the space of probability densities, together with a finite particle approximation based on Langevin dynamics. An error bound is provided, such that to reach an error $\epsilon$, the number of steps and the number of particles required are both polynomial in $1/\epsilon$.


**Limitations And Societal Impact:**

The authors adequately addressed the limitations of their work.

**Main Review:**

The paper makes a serious contribution to the growing literature on the global convergence of two layer neural networks, and is one of the first to provide a quantitative error bound, albeit under a somewhat more restrictive setting (requires entropy regularization with a large constant for the bound to be meaningful).
The paper is well-written and an effort is made to make it accessible to a broader audience, and this effort is commendable.

Perhaps because of this, most of the technical contribution is unfortunately pushed to the appendix, and there is no explicit error bound for PDA (the closest is Remark line 782 in the appendix). In other words, the main results (Theorem 1 and Corollary 2) concern an idealized mean field version (Algorithm 2) that is not implementable. I think it would be much better to write an explicit bound for PDA in the main paper.

On the exponential dependence on 1/\lambda_2: thank you for acknowledging this limitation and pointing to precisely why it arises. In the comment line 243, it is claimed that this dependence "is unfortunately unavoidable", is this alluding to a lower bound? If so, please include a reference, otherwise, please rephrase to make it less factual (for example "we believe that...") and provide a reasoning why this might be.

An additional point to clarify: line 665, it is mentioned that when considering the warm-start scheme, $h_x$ is set to $h_q$ ... without tolerance ($\epsilon = 0$). Why is it fine to assume $\epsilon = 0$ in this case? Even in this case one only gets a particle approximation of  q. Please clarify.

Additional suggestions:
- Consider adding a remark that the Gibbs distributions (such as $q_*^{(t)}$) are well defined thanks to L2 regularization and assumption (A1). This will clarify some of the importance of regularization.
- Appendix A: clarify what definition of gradient you are using, since $P_2$ is not a vector space.
- Overloaded notation: p denotes a density and the dimension of the parameter space (e.g. Definition A)
- Line 647: typo.
- Line 680: optimal coupling is not defined.
- The second and third experiments in the main paper don't seem to relate to the main narrative. They may be moved to the appendix to make additional space for an explicit bound on PDA.
- Experiment G.1: when comparing to SGD, it seems appropriate to also use entropy regularization (i.e. compare to Langevin gradient descent).

**Time Spent Reviewing:**

15

---

> ### Author Response · Authors · 2021-08-10
> **Response to Reviewer gSde**
>
> Thank you for thorough reading and helpful comments. Following your suggestion, we will work on the organization, add additional remarks and clarifications to improve the readability, and correct for typos and confusing notations.
> We address the technical comments below.
>
> - **"exponential dependence on $1/\lambda_2$ is unavoidable... is this alluding to a lower bound?"**
>
> Thank you for bringing this up. We made this comment due to existence of potential functions for which the Langevin algorithm could take exponential time to escape from local minima. For instance, for the famous double-well potential, Kramer's law [Berglund 2011] predicts that the expected time to transition out of one local basin is exponential in the height of the energy barrier. This implies slow convergence of the Langevin algorithm and also exponential dependence in the log-Sobolev constant (see Section 2.4 in [Menz and Schlichting 2012]).
>
> Consequently, unless we can rule out the existence of these "pathological" potential functions along the optimization trajectory *a priori*, it seems unlikely that such exponential factor can be avoided (as long as we are using the Langevin algorithm and convergence rate based on the log-Sobolev inequality). This is why we believe that additional assumption is likely required.
> This being said, we acknowledge that complexity *lower bound* for the Langevin algorithm has been a major open problem. We will rephrase this comment and make it less factual.
>
> Berglund 2011. *Kramers’ law: Validity, derivations and generalisations.*
>
> Menz and Schlichting 2012. *Poincare and logarithmic Sobolev inequalities by decomposition of the energy landscape.*
>
>
> - **"I think it would be much better to write an explicit bound for PDA in the main paper."**
>
> Thank you for the suggestion. We deferred the finite-particle analysis to the Appendix due to the space constraint, and also in order to keep the narrative simple. We will mention the finite-particle error in the main text in the revision.
>
>
> - **"it is mentioned that when considering the warm-start scheme, $h_x$ is set to $h_q$ ... without tolerance ($\epsilon=0$). Why is it fine to assume $\epsilon=0$ in this case? Even in this case one only gets a particle approximation of $q$. Please clarify."**
>
> Sorry for the confusion. The reason is that estimation of the approximation gap $|h_x^{(t)} - h_{q^{(t)}}(x_t) |$ is nontrivial for warm-start scheme, whereas for the resampling update it can be done by simply applying concentration inequalities.
> Hence, we utilized a different strategy for the finite-particle analysis for the warm-start update: we first prove the convergence of Algorithm 2, which corresponds to the case of $\epsilon = 0$, and then show that Algorithm 1 (the finite-particle version) converges to Algorithm 2 as $M$ becomes sufficiently large.

---

> > ### Comment · Reviewer_gSde · 2021-08-25
> > **Thank you for the clarifications**
> >
> > Thank you for the discussion and the references about dependence on $1/\lambda_2$, it would be great to see this discussion in the main body of the paper, as this is an important point readers should be aware of.
> > Thank you for the clarification about the $\epsilon = 0$ case.

---

> > > ### Author Response · Authors · 2021-08-25
> > > **Thank you for the followup.**
> > >
> > > Thank you for the additional comment. We will update our discussion on the dependence on $1/\lambda_2$ and incorporate the new references.

---

### Official Review · Reviewer_kvcd · 2021-07-16

**Rating:** 8
**Confidence:** 5

**Summary:**

The paper studies the convergence of the dual averaging method for the 1-hidden layer neural network in the mean-filed regime. Mean-field perspective allows one to recast original non-convex optimization problem over parameters of neural networks into (non necessarily strictly) convex problem in the space of probability measures.   Similarly to Mei et al. 2018 and Hu et al. 2019, authors consider entropy regularised cases under which the optimal solution to the infinite-dimensional problem over measures admits convenient Boltzman/Gibbs distribution.  Unlike Mei et al. 2018 and Hu et al. 2019,  who studied corresponding Langevin dynamics, while here authors study dual averaging methods in which one approximates the optimal sampling distribution by a sequence of simpler distributions that converge to it (with a rate one over a number of iterations), and use Langevin dynamics to sample from each distribution in the sequence.  If one would be able to quantify the rate of convergence of to each distribution in the sequence, this will imply the overall rate and complexity of the algorithm.

**Ethics Review Area:**

["I don’t know"]

**Main Review:**

Originality:  The idea of using a dual averaging algorithm (mainly used in online learning literature) is interesting. The convergence of the approximating sequence (outer loop) is established in Theorem 1, and this seems like the main contribution of this work.  Authors admit the proof idea is the same as in Nesterov 2009 and Xiao 2009, but I'm not familiar with these works.  What's puzzling is Assumption A3 that does not seem to be compatible with Theorem 2 in terms of rate. Furthermore, the authors never verify (at least not in Example 2 and Lemma B as they refer to) that each measure in the sequence satisfies log-Sobolev inequality (with what constant) and do not study the impact of the regularisation \lambda_1 on the rate of convergence.  Hu et al. already established an exponential rate of convergence in the case when \lambda_1 is sufficiently big.

In line 38 authors claim that continuous-time results cannot be easily translated to a discrete setting, but this is not true. See for example:
Jabir, J. F., Siska, D., & Szpruch, L. (2019). Mean-field neural odes via relaxed optimal control. arXiv preprint arXiv:1912.05475
for an example for such analysis in a fairly general setting. There is large body of research on uniform in time propagation of chaos and discretisation error for mean-field Langevin dynamics e.g:
- Majka, Mateusz B., Aleksandar Mijatović, and Lukasz Szpruch. "Nonasymptotic bounds for sampling algorithms without log-concavity." The Annals of Applied Probability 30.4 (2020): 1534-1581
- Delarue, François, and Alvin Tse. "Uniform in time weak propagation of chaos on the torus." arXiv preprint arXiv:2104.14973 (2021).

Clarity:  The presentation of the paper could improve in places. Some suggestions:
in section 2.1 the authors should stick to 1-hidden layer example. The ensemble of networks is confusing and not so interesting.
I struggled to gain intuition about what the dual averaging method is from section 3.1 and needed to go back to the literature to see what is going on.  Maybe authors could expand section 2.3 and explain dual averaging there.
Assumption A3 is in fact, a result that needs to be established. The authors should explain how this is done in Theorem 2

Significance:  Overall, this is an interesting research direction, but the presentation would need to improve, and a more thorough comparison with the literature needs to be presented i.e can authors can get away with less regularisation than Hu et al. and still obtain the rate of convergence?

**Time Spent Reviewing:**

3h

---

> ### Author Response · Authors · 2021-08-10
> **Response to Reviewer kvcd**
>
> Thank you for the detailed review and helpful comments.
>
> We first clarify a few important points that are potentially misunderstood in the review.
>
> - **"What's puzzling is Assumption A3 that does not seem to be compatible with Theorem 2..."**
>
> - **"The authors never verify (at least not in Example 2 and Lemma B as they refer to) that each measure in the sequence satisfies log-Sobolev inequality (with what constant)"**
>
> Assumption (A3) is compatible with Theorem 2 and the all the other statements in our study -- as commented on L.228-229, (A3) specifies the desired precision of the inner-loop problem, and decides the runtime of Langevin algorithm in Section 4.2.
> Moreover, the log-Sobolev constant of the intermediate sequence $q_\star^{(t)}$ is explicitly characterized (see below).
>
> We here give a brief recap of the proof.
> First, from the theory of dual averaging method we know that global convergence is guaranteed if $q^{(t)}$ provides a sufficient approximation to the Gibbs distribution $q_\star^{(t)}$.
> Assumption (A3) describes a required approximation accuracy for $q^{(t)}$; this is then achieved by running LMC to sample from $q^{(t)}_\star$, which always satisfies the log-Sobolev inequality under our regularized setting.
> We emphasize that the log-Sobolev inequality is needed only for $q^{(t)}_\star$, not for an inner-loop iterate $q^{(t)}$ (see Theorem 2).
> Indeed, $q^{(t)}_\star$ satisfies the log-Sobolev inequality with constant $\frac{2\lambda_1}{3\lambda_2 \exp(8/\lambda_2)}$ (see L.747 in Appendix).
> Note that this constant is verified by Example 2 and Lemma B because the potential function of $q^{(t)}_\star$ is the sum of bounded function and negative squared norm (i.e., strongly concave function) with coefficient $\frac{\lambda_1 (t-1)}{\lambda_2 (t+1)} \sim \frac{\lambda_1}{\lambda_2}$.
>
> - **"The authors ... do not study the impact of the regularisation $\lambda_1$ on the rate of convergence"**
>
> We highlight that our convergence theorem holds for *any* $\lambda_1 \geq 0$ and the dependency on $\lambda_1$ is also explicitly given.
> In particular, if we focus on the rate with respect to $\lambda_1$ and $T$ for simplicity, we obtain the total runtime complexity which is the product of the outer loop complexity $T=\tilde{O}( \max\\{ \lambda_1^{1/2} \epsilon^{-1/2}, \epsilon^{-1}\\} )$ and the maximal inner loop complexity $T_T = O\left( \frac{T^2(1+\lambda_1)^2}{\lambda_1 }\right) = \tilde{O}\left(\max\\{ \lambda_1\epsilon^{-1}, \epsilon^{-2} \\} \frac{(1+\lambda_1)^2}{\lambda_1} \right)$.
> For details see Theorem 1 and Corollary 2.
>
> We now comment on the relationship between our work and prior works mentioned in the review.
>
> - **"continuous-time results cannot be easily translated to a discrete setting, but this is not true. See for example: Jabir, J. F., Siska, D., and Szpruch, L. (2019)."**
>
> Thank you for bringing up several interesting works -- we will discuss them in our revision.
> We briefly remark on the difference between our study and each of these papers.
> -[Jabir et al.] analyzes a specific mean field dynamics and also its time discretization under appropriate conditions. However, to our knowledge,
> [Jabir et al.] requires strong regularization with sufficiently large coefficients $\lambda_1, \lambda_2$ to guarantee the convergence rate.
> In contrast, our convergence rate analysis holds for *any* $\lambda_1$ and $\lambda_2$, which we believe is an important contribution.
> -[Majka et al.] and [Delarue et al.] are also informative, though their analyses cannot be directly applied to our setting. Specifically, the dynamics in [Majka et al.] does not handle the interaction of particles, and our understanding is that [Delarue et al.] does not consider the time discretization; in addition, the problem setting in [Delarue et al.] may not cover the objective we consider.
>
> Also, we remark that converting continuous-time convergence rate into discrete-time result often requires extra care (e.g., continuous-time dynamics can be accelerated via rescaling time, but this does not imply accelerated convergence for the discretization [Wibisono et al. 2016]). Connecting the continuous-time analysis for mean field neural networks with the time and space discretization is an interesting direction.
>
> Wibisono et al. 2016. *A variational perspective on accelerated methods in optimization.*
>
>
>
> - **"[Hu et al.] already established an exponential rate of convergence in the case when $\lambda_1$ is sufficiently big."**
>
> We agree that [Hu et al.] also studied the convergence rate of the mean field Langevin dynamics.
> However, as commented above, one important distinction is that our proposed method can converge with a sublinear convergence rate for any $\lambda_1$, and we explicitly characterized the dependence on $\lambda_1$ (which turned out to be fairly mild); whereas in [Hu et al.], convergence rate for mean field Langevin dynamics is not given under small $\lambda_1$.
> Another important difference is that our proposed algorithm is purely discrete-time, so there is no continuous-time counterpart that needs to be discretized; this leads to a very different analysis and enables us to employ tools from the finite-dimensional optimization literature.
>
>
> - **"Authors admit the proof idea is the same as in Nesterov 2009 and Xiao 2009, but I'm not familiar with these works."**
>
> We remark that due to the difficulty of deriving convergence rate for neural networks in the mean field regime in general, extending the dual averaging method to infinite-dimensional optimization problems is also technically nontrivial and requires careful analysis.
> Hence we believe that designing learning algorithms for mean field neural network inspired by sophisticated finite dimensional optimization methods is an important contribution of our work.
>
>
> We would be happy to clarify any further concerns/questions in the discussion period.

---

> > ### Comment · Reviewer_kvcd · 2021-08-22
> > **Thank you for addressing my concerns.**
> >
> > Thank you for addressing my concerns.  It was a mistake on my end, and I've missed the LSI  constant for q_{\ast}^(t). Thanks for your clarification, and I'm happy to increase my score.
> >
> > About the comment of translating discrete-time finite particle limit: My remark was dictated by the author's comment in lines 37-38. All I was saying is that by carefully discretising, one often can preserve properties of the limiting system, and there are results in the literature that provide uniform in time bounds for both discretisation in time and space that can be adapted to various settings.
> >
> > I agree with the referee fmbs that a more thorough discussion on comparing the PDA and SGD would be welcome. For example, when working directly with SGD in one hidden layer setting in the mean filed regime, it appears that one needs more regularisation  (i.e large \lambda_1) to establish a quantitative convergence rate and LSI inequalities are tricky business.

---

> > > ### Author Response · Authors · 2021-08-25
> > > **Thank you for the followup.**
> > >
> > > Thank you for the updated review and helpful additional comments.
> > > We will highlight the required strong regularization in the current quantitative convergence analysis of mean field Langevin dynamics, and reword our statement in lines 37-38 regarding the discretization error.

---

### Decision · Program_Chairs · 2021-09-27

**Decision:**

Accept (Poster)

**Comment:**

The authors introduce a novel Particle Dual Averaging technique to minimize to minimize a non-linear functional of probability distributions. In terms of applications, they focus on the training of two-layer neural networks. In this context, they present quantitative convergence results for an entropy-regularized loss function. They also discuss the finite particle algorithm and present generalization bounds.

There was a short but informative discussion between reviewers. All the reviewers agree that the paper presents some original, interesting and solid theoretical contributions. However, it was felt by some reviewers that the experimental section was fairly weak and I encourage the authors to improve it before publication.

I recommend acceptance of the paper.